# ON SOCIALLY FAIR REGRESSION AND LOW-RANK APPROXIMATION

## ABSTRACT

Regression and low-rank approximation are two fundamental problems that are applied across a wealth of machine learning applications. In this paper, we study the question of socially fair regression and socially fair low-rank approximation, where the goal is to minimize the loss over all sub-populations of the data. We show that surprisingly, socially fair regression and socially fair low-rank approximation exhibit drastically different complexities. Specifically, we show that while fair regression can be solved up to arbitrary accuracy in polynomial time for a wide variety of loss functions, even constant-factor approximation to fair low-rank approximation requires exponential time under certain standard complexity hypotheses. On the positive side, we give an algorithm for fair low-rank approximation that, for a constant number of groups and constant-factor accuracy, runs in $2^{\mathrm{poly}(k)}$ rather than the naïve $n^{\mathrm{poly}(k)}$, which is a substantial improvement when the dataset has a large number $n$ of observations. Finally, we show that there exist bicriteria approximation algorithms for fair low-rank approximation and fair column subset selection that runs in polynomial time.

## 1 INTRODUCTION

Machine learning algorithms are increasingly used in technologies and decision-making processes that affect our daily lives, from high volume interactions such as online advertising, e-mail filtering, smart devices, or large language models, to more critical processes such as autonomous vehicles, healthcare diagnostics, credit scoring, and sentencing recommendations in courts of law (Chouldechova, 2017; Kleinberg et al., 2018; Berk et al., 2021). Machine learning algorithms frequently require statistical analysis, utilizing fundamental problems from numerical linear algebra, especially regression and low-rank approximation.

In the classical regression problem, the input is a data matrix $\mathbf{A} \in \mathbb{R}^{n \times d}$ and a label matrix $\mathbf{b} \in \mathbb{R}^n$, and the task is to determine the hidden vector $\mathbf{x} \in \mathbb{R}^d$ that minimizes $\mathcal{L}(\mathbf{Ax} - \mathbf{b})$ for some loss function $\mathcal{L}$. The vector $\mathbf{x}$ can then be subsequently used to label future observations $\mathbf{v}$ through the operation $\langle \mathbf{v}, \mathbf{x} \rangle$. Hence, regression analysis is frequently used in machine learning to infer causal relationships between the independent and dependent variables, and thus also used for prediction and forecasting. Similarly, in the classical low-rank approximation problem, the input is a data matrix $\mathbf{A} \in \mathbb{R}^{n \times d}$ and an integral rank parameter $k > 0$, and the goal is to find the best rank $k$ approximation to $\mathbf{A}$, i.e., finding a set of $k$ vectors in $\mathbb{R}^d$ that span a matrix $\mathbf{B}$, which minimizes $\mathcal{L}(\mathbf{A} - \mathbf{B})$ across all rank-$k$ matrices $\mathbf{B}$, for some loss function $\mathcal{L}$. The rank parameter $k$ should be chosen to accurately represent the complexity of the underlying model chosen to fit the data, and thus the low-rank approximation problem is often used for mathematical modeling and data compression. Regression and low-rank approximation share key connections and thus are often studied together, e.g., a number of sketching techniques can often be applied to both problems, resulting in significant runtime improvements due to dimensionality reduction. See Appendix A.1 for more information.

**Algorithmic fairness.** Unfortunately, real-world machine learning algorithms across a wide variety of domains have recently produced a number of undesirable outcomes from the lens of generalization. For example, Barocas & Selbst (2016) noted that decision-making processes using data collected from smartphone devices reporting poor road quality could potentially underserve poorer communities with less smartphone ownership. Kay et al. (2015) observed that search queries for CEOs overwhelmingly

returned images of white men, while Buolamwini & Gebru (2018) observed that facial recognition software exhibited different accuracy rates for white men compared with dark-skinned women.

Initial attempts to explain these issues can largely be categorized into either "biased data" or "biased algorithms", where the former might include training data that is significantly misrepresenting the true statistics of some sub-population, while the latter might sacrifice accuracy on a specific sub-population in order to achieve better global accuracy. As a result, an increasingly relevant line of active work has focused on designing *fair* algorithms. An immediate challenge is to formally define the desiderata demanded from fair algorithmic design and indeed multiple natural quantitative measures of fairness have been proposed (Hardt et al., 2016; Chouldechova, 2017; Kleinberg et al., 2017; Berk et al., 2021). However, Kleinberg et al. (2017); Corbett-Davies et al. (2017) showed that many of these conditions for fairness cannot be simultaneously achieved.

In this paper, we focus on *socially fair* algorithms, which seek to optimize the performance of the algorithm across all sub-populations. That is, for the purposes of regression and low-rank approximation, the goal is to minimize the maximum cost across all sub-populations. For socially fair regression, the input is a set of data matrices $\mathbf{A}^{(1)} \in \mathbb{R}^{n_1 \times d}, \ldots, \mathbf{A}^{(\ell)} \in \mathbb{R}^{n_\ell \times d}$ and labels $\mathbf{b}^{(1)} \in \mathbb{R}^{n_1}, \ldots, \mathbf{b}^{(\ell)} \in \mathbb{R}^{n_\ell}$ corresponding to $\ell$ sub-populations, which we refer to as groups. The goal is to optimize the objective function $\min_{\mathbf{x} \in \mathbb{R}^d} \max_{i \in [\ell]} \|\mathbf{A}^{(i)}\mathbf{x} - \mathbf{b}^{(i)}\|$, for some fixed norm $\|\cdot\|$. Similarly, for socially fair low-rank approximation, the input is a set of data matrices $\mathbf{A}^{(1)} \in \mathbb{R}^{n_1 \times d}, \ldots, \mathbf{A}^{(\ell)} \in \mathbb{R}^{n_\ell \times d}$ corresponding to $\ell$ groups and a rank parameter $k$, and the goal is to determine a set of $k$ factors $\mathbf{U}_1, \ldots, \mathbf{U}_k \in \mathbb{R}^d$ that span matrices $\mathbf{B}^{(1)}, \ldots, \mathbf{B}^{(\ell)}$, which minimize $\max_{i \in [\ell]} \|\mathbf{A}^{(i)} - \mathbf{B}^{(i)}\|_F$. Due to the Eckart–Young–Mirsky theorem stating that the Frobenius loss is minimized when each $\mathbf{A}^{(i)}$ is projected onto the span of $\mathbf{U} = \mathbf{U}_1 \circ \ldots \circ \mathbf{U}_k$, the problem is equivalent to $\min_{\mathbf{U} \in \mathbb{R}^{k \times d}} \max_{i \in [\ell]} \|\mathbf{A}^{(i)}\mathbf{U}^\dagger\mathbf{U} - \mathbf{A}^{(i)}\|_F$, where $\dagger$ denotes the Moore-Penrose pseudoinverse.

## 1.1 OUR CONTRIBUTIONS AND TECHNICAL OVERVIEW

Surprisingly, we show in this paper that socially fair regression and socially fair low-rank approximation exhibit drastically different complexities. Specifically, we show that while fair regression can be solved up to arbitrary accuracy in polynomial time for a wide variety of loss functions, even constant-factor approximation to fair low-rank approximation requires exponential time under certain standard complexity hypotheses.

**Fair regression.** We first summarize a number of simple results for socially fair regression. Using known connections to convex minimization Abernethy et al. (2022), we show that given an accuracy parameter $\varepsilon > 0$, fair regression can be solved up to additive error $\varepsilon$ in polynomial time for any convex loss function whose subgradient can also be computed in polynomial time.

**Theorem 1.1.** *Let $n_1, \ldots, n_\ell$ be positive integers and for each $i \in [\ell]$, let $\mathbf{A}^{(i)} \in \mathbb{R}^{n_i \times d}$ and $\mathbf{b}^{(i)} \in \mathbb{R}^{n_i}$. Let $\Delta = \mathrm{poly}(n)$ for $n = n_1 + \ldots + n_\ell$ and let $\varepsilon \in (0, 1)$. For any norm $\|\cdot\|$ whose subgradient can be computed in $\mathrm{poly}(n, d)$ time, there exists an algorithm that outputs $\mathbf{x}^* \in [-\Delta, \Delta]^d$ such that $\max_{i \in [\ell]} \|\mathbf{A}^{(i)}\mathbf{x}^* - \mathbf{b}^{(i)}\| \leq \varepsilon + \min_{\mathbf{x} \in [-\Delta, \Delta]^d} \max_{i \in [\ell]} \|\mathbf{A}^{(i)}\mathbf{x} - \mathbf{b}^{(i)}\|$. The algorithm uses $\mathrm{poly}\left(n, d, \frac{1}{\varepsilon}\right)$ runtime.*

Importantly, the class of convex loss functions whose subgradient can be computed in polynomial time include the popular class of $L_p$ norms.

The algorithm corresponding to the guarantees of Theorem 1.1 is quite simple. Recall that the maximum of convex functions is itself a convex function. Thus if the loss function is convex, then the fair regression problem corresponds to minimizing a convex objective over a convex space. Abernethy et al. (2022) uses this observation in conjunction with stochastic projection gradient descent. We instead apply existing results in convex optimization stating that the objective can be efficiently approximated to within additive error $\varepsilon$ given a separation oracle (Lee et al., 2015; Jiang et al., 2020b). Moreover, Lee et al. (2015) notes that such a separation oracle can be implemented using the subgradient of the loss function and the separating hyperplane method.

In many cases, the input matrices $\mathbf{A}^{(1)}, \ldots, \mathbf{A}^{(\ell)}$ can be somewhat sparse. In these classes, the number of nonzero entries in the matrix $\mathbf{A} = \mathbf{A}^{(1)} \circ \ldots \circ \mathbf{A}^{(\ell)}$, which we denote by $\mathrm{nnz}(\mathbf{A})$, can be significantly smaller than $nd$. We also give a result for fair regression that uses input-sparsity runtime.

**Theorem 1.2.** *Let $n_1, \ldots, n_\ell$ be positive integers and for each $i \in [\ell]$, let $\mathbf{A}^{(i)} \in \mathbb{R}^{n_i \times d}$ and $\mathbf{b}^{(i)} \in \mathbb{R}^{n_i}$. Let $\Delta = \mathrm{poly}(n)$ for $n = n_1 + \ldots + n_\ell$ and let $\varepsilon \in (0, 1)$. There exists an algorithm that outputs $\mathbf{x}^* \in [-\Delta, \Delta]^d$ such that $\max_{i \in [\ell]} \|\mathbf{A}^{(i)} \mathbf{x}^* - \mathbf{b}^{(i)}\| \leq (1 + \varepsilon) \min_{\mathbf{x} \in [-\Delta, \Delta]^d} \max_{i \in [\ell]} \|\mathbf{A}^{(i)} \mathbf{x} - \mathbf{b}^{(i)}\|$. The algorithm uses $\tilde{\mathcal{O}}\left(\mathrm{nnz}(\mathbf{A})\right) + \mathrm{poly}\left(d, \ell, \frac{1}{\varepsilon}\right)$ runtime.*

We achieve Theorem 1.2 by applying existing results in subspace embeddings that use input-sparsity runtime to first reduce the overall dimension of the entire dataset, and then we apply the algorithm of Theorem 1.1.

**Fair low-rank approximation.** We now describe our results for socially fair low-rank approximation. We first show that under assumption that $\mathrm{P} \neq \mathrm{NP}$, fair low-rank approximation cannot be approximated within any constant factor in polynomial time.

**Theorem 1.3.** *Fair low-rank approximation is NP-hard to approximate within any constant factor.*

We show Theorem 1.3 by reducing to the problem of minimizing the distance of a set of $n$ points in $d$-dimensional Euclidean space to all set of $k$ dimensional linear subspaces, which was shown by (Brieden et al., 2000; Deshpande et al., 2011) to be NP-hard to approximate within any constant factor. In fact, Brieden et al. (2000); Deshpande et al. (2011) showed that a constant-factor approximation to this problem requires runtime exponential in $k$ under a stronger assumption, the exponential time hypothesis (Impagliazzo & Paturi, 2001). We show similar results for the fair low-rank approximation problem.

**Theorem 1.4.** *Under the exponential time hypothesis, the fair low-rank approximation requires $2^{k^{\Omega(1)}}$ time to approximate within any constant factor.*

Together, Theorem 1.3 and Theorem 1.4 show that under standard complexity assumptions, we cannot achieve constant-factor approximation to fair low-rank approximation using time polynomial in $n$ and exponential in $k$. We thus consider additional relaxations, such as bicriteria approximation (Theorem 1.6) or $2^{\mathrm{poly}(k)}$ runtime (Theorem 1.5). On the positive side, we first show that for a constant number of groups and constant-factor accuracy, it suffices to use runtime $2^{\mathrm{poly}(k)}$ rather than the naïve $n^{\mathrm{poly}(k)}$, which is a substantial improvement when the dataset has a large number of observations, i.e., $n$ is large.

**Theorem 1.5.** *Given an accuracy parameter $\varepsilon \in (0, 1)$, there exists an algorithm that outputs $\widetilde{\mathbf{V}} \in \mathbb{R}^{k \times d}$ such that with probability at least $\frac{2}{3}$, $\max_{i \in [\ell]} \|\mathbf{A}^{(i)} (\widetilde{\mathbf{V}})^\dagger \widetilde{\mathbf{V}} - \mathbf{A}^{(i)}\|_F \leq (1 + \varepsilon) \cdot \min_{\mathbf{V} \in \mathbb{R}^{k \times d}} \max_{i \in [\ell]} \|\mathbf{A}^{(i)} \mathbf{V}^\dagger \mathbf{V} - \mathbf{A}^{(i)}\|_F$. The algorithm uses runtime $\frac{1}{\varepsilon} \mathrm{poly}(n) \cdot (2\ell)^{\mathcal{O}(N)}$, for $n = \sum_{i=1}^{\ell} n_i$ and $N = \mathrm{poly}\left(\ell, k, \frac{1}{\varepsilon}\right)$.*

At a high level, the algorithm corresponding to Theorem 1.5 first finds a value $\alpha$ such that is an $\ell$-approximation to the optimal solution. It then repeatedly decreases $\alpha$ by $(1 + \varepsilon)$ factors while checking if the resulting quantity is still feasible. To efficiently check whether $\alpha$ is feasible, we first reduce the dimension of the data using an affine embedding matrix $\mathbf{S}$, so that for all rank $k$ matrices $\mathbf{V}$ and all $i \in [\ell]$, $(1 - \varepsilon) \|\mathbf{A}^{(i)} \mathbf{V}^\dagger \mathbf{V} - \mathbf{A}^{(i)}\|_F^2 \leq \|\mathbf{A}^{(i)} \mathbf{V}^\dagger \mathbf{V} \mathbf{S} - \mathbf{A}^{(i)} \mathbf{S}\|_F^2 \leq (1 + \varepsilon) \|\mathbf{A}^{(i)} \mathbf{V}^\dagger \mathbf{V} - \mathbf{A}^{(i)}\|_F^2$.

Observe that given $\mathbf{V}$, it is known through the closed form solution to the regression problem that the rank-$k$ minimizer of $\|\mathbf{X}^{(i)} \mathbf{V} \mathbf{S} - \mathbf{A}^{(i)} \mathbf{S}\|_F^2$ is $(\mathbf{A}^{(i)} \mathbf{S})(\mathbf{V} \mathbf{S})^\dagger$. Let $\mathbf{R}^{(i)}$ be defined so that $(\mathbf{A}^{(i)} \mathbf{S})(\mathbf{V} \mathbf{S})^\dagger \mathbf{R}^{(i)}$ has orthonormal columns, so that

$$\|(\mathbf{A}^{(i)} \mathbf{S})(\mathbf{V} \mathbf{S})^\dagger \mathbf{R}^{(i)})((\mathbf{A}^{(i)} \mathbf{S})(\mathbf{V} \mathbf{S})^\dagger \mathbf{R}^{(i)})^\dagger \mathbf{A}^{(i)} \mathbf{S} - \mathbf{A}^{(i)} \mathbf{S}\|_F^2 = \min_{\mathbf{X}^{(i)}} \|\mathbf{X}^{(i)} \mathbf{V} \mathbf{S} - \mathbf{A}^{(i)} \mathbf{S}\|_F^2.$$

It follows that if $\alpha$ is feasible, then $\alpha(1 + \varepsilon) \geq \|(\mathbf{A}^{(i)} \mathbf{S})(\mathbf{V} \mathbf{S})^\dagger \mathbf{R}^{(i)})((\mathbf{A}^{(i)} \mathbf{S})(\mathbf{V} \mathbf{S})^\dagger \mathbf{R}^{(i)})^\dagger \mathbf{A}^{(i)} \mathbf{S} - \mathbf{A}^{(i)} \mathbf{S}\|_F^2$.

Unfortunately, $\mathbf{V}$ is not given, so the above approach will not quite work. Instead, we use a polynomial solver to check whether there exists such a $\mathbf{V}$ by writing $\mathbf{Y} = \mathbf{V} \mathbf{S}$ and its pseudoinverse $\mathbf{W} = (\mathbf{V} \mathbf{S})^\dagger$ and check whether there exists a satisfying assignment to the above inequality, given the constraints (1) $\mathbf{Y} \mathbf{W} \mathbf{Y} = \mathbf{Y}$, (2) $\mathbf{W} \mathbf{Y} \mathbf{W} = \mathbf{W}$, and (3) $\mathbf{A}^{(i)} \mathbf{S} \mathbf{W} \mathbf{R}^{(i)}$ has orthonormal columns.

We remark that because $\mathbf{V} \in \mathbb{R}^{k \times d}$, then we cannot naïvely implement the polynomial system, because it would require $kd$ variables and thus use $2^{\Omega(dk)}$ runtime. Instead, we only manipulate $\mathbf{V} \mathbf{S}$,

which has dimension $k \times m$ for $m = \mathcal{O}\left(\frac{k^2}{\varepsilon^2}\log \ell\right)$, allowing the polynomial system solver to achieve $2^{\text{poly}(mk)}$ runtime.

Next, we show that there exists a bicriteria approximation algorithm for fair low-rank approximation that uses polynomial runtime.

**Theorem 1.6.** *Given a trade-off parameter $c \in (0,1)$, there exists an algorithm that outputs $\widetilde{\mathbf{V}} \in \mathbb{R}^{t \times d}$ for $t = \mathcal{O}\left(k(\log\log k)(\log^2 d)\right)$ such that with probability at least $\frac{2}{3}$,*

$$\max_{i \in [\ell]} \|\mathbf{A}^{(i)}(\widetilde{\mathbf{V}})^\dagger \widetilde{\mathbf{V}} - \mathbf{A}^{(i)}\|_F \leq \ell^c \cdot 2^{1/c} \cdot \mathcal{O}\left(k(\log\log k)(\log d)\right) \min_{\mathbf{V} \in \mathbb{R}^{k \times d}} \max_{i \in [\ell]} \|\mathbf{A}^{(i)}\mathbf{V}^\dagger \mathbf{V} - \mathbf{A}^{(i)}\|_F.$$

*The algorithm uses runtime polynomial in $n$ and $d$.*

The algorithm for Theorem 1.6 substantially differs from that of Theorem 1.5. For one, we can no longer use a polynomial system solver, because it would be infeasible to achieve polynomial runtime. Instead, we observe that for sufficiently large $p$, we have $\max \|\mathbf{x}\|_\infty = (1 \pm \varepsilon)\|\mathbf{x}\|_p$ and thus focus on optimizing $\min_{\mathbf{V} \in \mathbb{R}^{k \times d}} \left(\sum_{i \in [\ell]} \|\mathbf{A}^{(i)}\mathbf{V}^\dagger\mathbf{V} - \mathbf{A}^{(i)}\|_F^p\right)^{1/p}$. However, the terms $\|\mathbf{A}^{(i)}\mathbf{V}^\dagger\mathbf{V} - \mathbf{A}^{(i)}\|_F^p$ are difficult to handle, so we apply Dvoretzky's Theorem, i.e., Theorem 3.2, to generate matrices $\mathbf{G}$ and $\mathbf{H}$ so that $(1 - \varepsilon)\|\mathbf{GMH}\|_p \leq \|\mathbf{M}\|_F \leq (1 + \varepsilon)\|\mathbf{GMH}\|_p$, for all matrices $\mathbf{M} \in \mathbb{R}^{n \times d}$, so that it suffices to approximately solve $\min_{\mathbf{X} \in \mathbb{R}^{k \times d}} \|\mathbf{GAHSX} - \mathbf{GAH}\|_p$, for $\mathbf{A} = \mathbf{A}^{(1)} \circ \ldots \circ \mathbf{A}^{(\ell)}$.

Although low-rank approximation with $L_p$ loss cannot be well-approximated in polynomial time, we recall that there exists a matrix $\mathbf{S}$ that samples a "small" number of columns of $\mathbf{A}$ to provide a coarse bicriteria approximation to $L_p$ low-rank approximation (Woodruff & Yasuda, 2023a). However, we require a solution with dimension $d$ and thus we seek to solve regression problem $\min_{\mathbf{X}} \|\mathbf{GAHSX} - \mathbf{GAH}\|_p$. Thus, we consider a Lewis weight sampling matrix $\mathbf{T}$ such that

$$\frac{1}{2}\|\mathbf{TGAHSX} - \mathbf{TGAH}\|_p \leq \|\mathbf{GAHSX} - \mathbf{GAH}\|_p \leq 2\|\mathbf{TGAHSX} - \mathbf{TGAH}\|_p.$$

and again note that $(\mathbf{TGAHS})^\dagger \mathbf{TGAH}$ is the closed-form solution to the minimization problem $\min_{\mathbf{x}} \|\mathbf{TGAHSX} - \mathbf{TGAH}\|_F$, which only provides a small distortion to the $L_p$ regression problem, since $\mathbf{TGAH}$ has a small number of rows due to the dimensionality reduction. We then observe that by Dvoretzky's Theorem, $(\mathbf{TGAHS})^\dagger \mathbf{TGA}$ is a "good" approximate solution to the original fair low-rank approximation problem. Given $\delta \in (0,1)$, the success probabilities for both Theorem 1.5 and Theorem 1.6 can be boosted to arbitrary $1 - \delta$ by taking the minimum of $\mathcal{O}\left(\log\frac{1}{\delta}\right)$ independent instances of the algorithm, at the cost of increasing the runtime by the same factor.

Although low-rank approximation can reveal important latent structure among the dataset, the resulting linear combinations may not be as interpretable as simply selecting $k$ features. The column subset selection problem is therefore a version of low-rank approximation with the restriction that the right factor must be $k$ columns of the data matrix. We give a bicriteria approximation algorithm for fair column subset selection that uses polynomial runtime.

**Theorem 1.7.** *Given input matrices $\mathbf{A}^{(i)} \in \mathbb{R}^{n_i \times d}$ with $n = \sum n_i$, there exists an algorithm that selects a set $S$ of $k' = \mathcal{O}\left(k \log k\right)$ columns such that with probability at least $\frac{2}{3}$, $S$ is a $\mathcal{O}\left(k(\log\log k)(\log d)\right)$-approximation to the fair column subset selection problem. The algorithm uses runtime polynomial in $n$ and $d$.*

The immediate challenge in adapting the previous approach for fair low-rank approximation to fair column subset selection is that we required the Gaussian matrices $\mathbf{G}, \mathbf{H}$ to embed the awkward maximum of Frobenius losses $\min \max_{i \in [\ell]} \|\cdot\|_F$ into the more manageable $L_p$ loss $\min \|\cdot\|_p$ through $\mathbf{GAH}$. However, selecting columns of $\mathbf{GAH}$ does not correspond to selecting columns of the input matrices $\mathbf{A}^{(1)}, \ldots, \mathbf{A}^{(\ell)}$.

Instead, we view the bicriteria solution $\widetilde{\mathbf{V}}$ from fair low-rank approximation as a good starting point for the right factor for fair low-rank approximation. Thus we consider the multi-response regression problem $\max_{i \in [\ell]} \min_{\mathbf{B}^{(i)}} \|\mathbf{B}^{(i)}\widetilde{\mathbf{V}} - \mathbf{A}^{(i)}\|_F$. We then argue through Dvoretzky's theorem that a leverage score sampling matrix $\mathbf{S}$ that samples $\mathcal{O}\left(k \log k\right)$ columns of $\widetilde{\mathbf{V}}$ will provide a good approximation to the column subset selection problem. We defer the formal exposition to Appendix D.

**Empirical evaluations.** Finally, in Section 4, we present a number of experimental results on socially fair regression, comparing the performance of the socially fair objective values associated with the outputs of the fair regression algorithm and the standard regression algorithms.

Our experimental results in Figure 3 similarly demonstrate the improvement of the fair regression algorithm over the standard regression algorithm across various values of $k$ and various matrix sizes. We perform our experiments on both synthetic data and the Law School Admissions Councils National Longitudinal Bar Passage Study (Wightman, 1998); in the latter dataset, the goal is to predict a student's first year GPA at law school via least squared regression, given the student's undergraduate GPA and whether or not they passed the bar exam, using race as the protected attribute. Our results demonstrate the improvement of the fair regression algorithm over the standard regression algorithm across various datasets, number of features, number of observations, and number of protected groups. Finally, we give a simple synthetic example for socially fair low-rank approximation that serves as a proof-of-concept similarly demonstrating the improvement of fair low-rank approximation optimal solutions over the optimal solutions for standard low-rank approximation.

## 1.2 RELATED WORK

Initial insight into *socially fair data summarization* methods were presented by Samadi et al. (2018), where the concept of *fair PCA* was explored. This study introduced the fairness metric of average reconstruction loss, expressed by the loss function $\text{loss}(\mathbf{A}, \mathbf{B}) := \|\mathbf{A} - \mathbf{B}\|_F^2 - \|\mathbf{A} - \mathbf{A}_k\|_F^2$, aiming to identify a $k$-dimensional subspace that minimizes the loss across the groups, with $\mathbf{A}_k$ representing the best rank-$k$ approximation of $\mathbf{A}$. Their proposed approach, in a two-group scenario, identifies a fair PCA of up to $k + 1$ dimensions that is not worse than the optimal fair PCA with $k$ dimensions. When extended to $\ell$ groups, this method requires an additional $k + \ell - 1$ dimensions. Subsequently, Tantipongpipat et al. (2019) explored fair PCA from a distinct objective perspective, seeking a projection matrix $\mathbf{P}$ optimizing $\min_{i \in [\ell]} \|\mathbf{A}^{(i)}\mathbf{P}\|_F^2$. A pivotal difference between these works and ours is our focus on the reconstruction error objective, a widely accepted objective for regression and low-rank approximation tasks. Alternatively, Olfat & Aswani (2019); Lee et al. (2022); Kleindessner et al. (2023) explored a different formulation of fair PCA. The main objective is to ensure that data representations are not influenced by demographic attributes. In particular, when a classifier is exposed only to the projection of points onto the $k$-dimensional subspace, it should be unable to predict the demographic attributes.

For fair regression, initial research focused on designing models that offer similar treatment to instances with comparable observed results by incorporating *fairness regularizers* Berk et al. (2017). However, in Agarwal et al. (2019), the authors studied a fairness notion closer to our optimization problem, termed as "bounded group loss". In their work, the aim is to cap each group's loss within a specific limit while also optimizing the cumulative loss. Notably, their approach diverged from ours, with a focus on the sample complexity and the problem's generalization error bounds.

Abernethy et al. (2022) studied a similar socially fair regression problem under the name min-max regression. In their setting, the goal is to minimize the maximum loss over a mixture distribution, given samples from the mixture; our fair regression setting can be reduced to theirs. Abernethy et al. (2022) observed that a maximum of norms is a convex function and can therefore be solved using projected stochastic gradient descent.

The term "socially fair" was first introduced in the context of clustering, aiming to optimize clustering costs across predefined group sets (Ghadiri et al., 2021; Abbasi et al., 2021). In subsequent studies, tight approximation algorithms (Makarychev & Vakilian, 2021; Chlamtáč et al., 2022), FPT approaches (Goyal & Jaiswal, 2023), and bicriteria approximation algorithms (Ghadiri et al., 2022) for socially fair clustering have been presented.

## 2 SOCIALLY FAIR REGRESSION

As a warm-up, we first present simple and intuitive algorithms for fair regression. Let $\ell$ be the number of groups, $n_1, \ldots, n_\ell$ be positive integers and for each $i \in [\ell]$, let $\mathbf{A}^{(i)} \in \mathbb{R}^{n_i \times d}$ and $\mathbf{b}^{(i)} \in \mathbb{R}^{n_i}$. Then for a norm $\|\cdot\|$, we define the fair regression problem to be $\min_{\mathbf{x} \in \mathbb{R}^d} \max_{i \in [\ell]} \|\mathbf{A}^{(i)}\mathbf{x} - \mathbf{b}^{(i)}\|$. We first show that the optimal solution to the standard regression minimization problem also admits a $\ell$-approximation to the fair regression problem.

**Theorem 2.1.** *The optimal solution to the standard regression problem that computes a vector $\widehat{\mathbf{x}} \in \mathbb{R}^d$ also satisfies $\max_{i \in [\ell]} \|\mathbf{A}^{(i)}\widehat{\mathbf{x}} - \mathbf{b}^{(i)}\|_2 \le \ell \cdot \min_{\mathbf{x} \in \mathbb{R}^d} \max_{i \in [\ell]} \|\mathbf{A}^{(i)}\mathbf{x} - \mathbf{b}^{(i)}\|_2$, i.e., the algorithm outputs a $\ell$-approximation to the fair regression problem. For $L_2$ loss, the algorithm uses $\mathcal{O}\left(nd^{\omega-1}\right)$ runtime, where $n = n_1 + \ldots + n_\ell$ and $\omega$ is the matrix multiplication exponent.*

More generally, we can apply the same principles to observe that for any norm $\|\cdot\|$, $\|\mathbf{A}\mathbf{x}^* - \mathbf{b}\| \le \sum_{i=1}^{\ell} \|\mathbf{A}^{(i)}\mathbf{x}^* - \mathbf{b}^{(i)}\| \le \ell \cdot \max_{i \in [\ell]} \|\mathbf{A}^{(i)}\mathbf{x}^* - \mathbf{b}^{(i)}\|$. Thus if $\|\cdot\|$ admits an efficient algorithm for regression, then $\|\cdot\|$ also admits an efficient $\ell$-approximation algorithm for fair regression. See Algorithm 1 for reference.

We now give a general algorithm for achieving additive $\varepsilon$ approximation to fair regression. Abernethy et al. (2022) observed that every norm is convex, since $\|\lambda \mathbf{u} + (1-\lambda)\mathbf{v}\| \le \lambda\|\mathbf{u}\| + (1-\lambda)\|\mathbf{v}\|$ by triangle inequality. Therefore, the function $g(\mathbf{x}) := \max_{i \in [\ell]}\{\|\mathbf{A}^{(i)}\mathbf{x} - \mathbf{b}^{(i)}\|\}$ is convex because the maximum of convex functions is also a convex function. Hence, the objective $\min_{\mathbf{x}} g(\mathbf{x})$ is the minimization of a convex function and can be solved using standard tools in convex optimization. Abernethy et al. (2022) leveraged this observation by applying projected stochastic gradient descent. Here we use a convex solvers based on separating hyperplanes. The resulting algorithm is quite simple and appears in Algorithm 2. We defer the proof to Appendix B.2.

---

**Algorithm 1** $\ell$-approximation for Socially Fair Regression

**Input:** $\mathbf{A}^{(i)} \in \mathbb{R}^{n_i \times d}$, $\mathbf{b}^{(i)} \in \mathbb{R}^{n_i}$ for all $i \in [\ell]$

**Output:** $\ell$-approximation for fair regression
1: $\mathbf{A} \leftarrow \mathbf{A}^{(1)} \circ \ldots \circ \mathbf{A}^{(\ell)}$
2: $\mathbf{b} \leftarrow \mathbf{A}^{(1)} \circ \ldots \circ \mathbf{b}^{(\ell)}$
3: **return** $\operatorname{argmin}_{\mathbf{x} \in \mathbb{R}^d} \|\mathbf{A}\mathbf{x} - \mathbf{b}\|$

---

**Algorithm 2** Algorithm for Socially Fair Regression

**Input:** $\mathbf{A}^{(i)} \in \mathbb{R}^{n_i \times d}$, $\mathbf{b}^{(i)} \in \mathbb{R}^{n_i}$ for all $i \in [\ell]$

**Output:** Optimal for socially fair regression
1: Use a convex solver to return $\operatorname{argmin}_{\mathbf{x} \in \mathbb{R}^d} \max_{i \in [\ell]} \|\mathbf{A}^{(i)}\mathbf{x} - \mathbf{b}^{(i)}\|$

---

## 3 SOCIALLY FAIR LOW-RANK APPROXIMATION

In this section, we consider algorithms and hardness for socially fair low-rank approximation. Let $n_1, \ldots, n_\ell$ be positive integers and for each $i \in [\ell]$, let $\mathbf{A}^{(i)} \in \mathbb{R}^{n_i \times d}$. Then for a norm $\|\cdot\|$, we define the fair low-rank approximation problem to be $\min_{\mathbf{V} \in \mathbb{R}^{k \times d}} \max_{i \in [\ell]} \|\mathbf{A}^{(i)}\mathbf{V}^\top\mathbf{V} - \mathbf{A}^{(i)}\|$.

### 3.1 $(1+\varepsilon)$-APPROXIMATION ALGORITHM FOR FAIR LOW-RANK APPROXIMATION

We first give a $(1+\varepsilon)$-approximation algorithm for fair low-rank approximation that uses runtime $\frac{1}{\varepsilon}\operatorname{poly}(n) \cdot (2\ell)^{\mathcal{O}(N)}$, for $n = \sum_{i=1}^{\ell} n_i$ and $N = \operatorname{poly}\left(\ell, k, \frac{1}{\varepsilon}\right)$.

The algorithm first finds a value $\alpha$ that is an $\ell$-approximation to the optimal solution, i.e.,

$$\min_{\mathbf{V} \in \mathbb{R}^{k \times d}} \max_{i \in [\ell]} \|\mathbf{A}^{(i)}\mathbf{V}^\dagger\mathbf{V} - \mathbf{A}^{(i)}\|_F \le \alpha \le \ell \cdot \min_{\mathbf{V} \in \mathbb{R}^{k \times d}} \max_{i \in [\ell]} \|\mathbf{A}^{(i)}\mathbf{V}^\dagger\mathbf{V} - \mathbf{A}^{(i)}\|_F.$$

We then repeatedly decrease $\alpha$ by $(1+\varepsilon)$ while checking if the resulting quantity is still achievable. To efficiently check if $\alpha$ is achievable, we first apply dimensionality reduction to each of the matrices by right-multiplying by an affine embedding matrix $\mathbf{S}$, so that

$$(1-\varepsilon)\|\mathbf{A}^{(i)}\mathbf{V}^\dagger\mathbf{V} - \mathbf{A}^{(i)}\|_F^2 \le \|\mathbf{A}^{(i)}\mathbf{V}^\dagger\mathbf{V}\mathbf{S} - \mathbf{A}^{(i)}\mathbf{S}\|_F^2 \le (1+\varepsilon)\|\mathbf{A}^{(i)}\mathbf{V}^\dagger\mathbf{V} - \mathbf{A}^{(i)}\|_F^2,$$

for all rank $k$ matrices $\mathbf{V}$ and all $i \in [\ell]$.

Now if we knew $\mathbf{V}$, then for each $i \in [\ell]$, we can find $\mathbf{X}^{(i)}$ that minimizes $\|\mathbf{X}^{(i)}\mathbf{V}\mathbf{S} - \mathbf{A}^{(i)}\mathbf{S}\|_F^2$ and the resulting quantity will approximate $\|\mathbf{A}^{(i)}\mathbf{V}^\dagger\mathbf{V} - \mathbf{A}^{(i)}\|_F^2$. In fact, we know that the minimizer is $(\mathbf{A}^{(i)}\mathbf{S})(\mathbf{V}\mathbf{S})^\dagger$ through the closed form solution to the regression problem. Let $\mathbf{R}^{(i)}$ be defined so that $(\mathbf{A}^{(i)}\mathbf{S})(\mathbf{V}\mathbf{S})^\dagger\mathbf{R}^{(i)}$ has orthonormal columns, so that

$$\|(\mathbf{A}^{(i)}\mathbf{S})(\mathbf{V}\mathbf{S})^\dagger\mathbf{R}^{(i)})((\mathbf{A}^{(i)}\mathbf{S})(\mathbf{V}\mathbf{S})^\dagger\mathbf{R}^{(i)})^\dagger\mathbf{A}^{(i)}\mathbf{S} - \mathbf{A}^{(i)}\mathbf{S}\|_F^2 = \min_{\mathbf{X}^{(i)}} \|\mathbf{X}^{(i)}\mathbf{V}\mathbf{S} - \mathbf{A}^{(i)}\mathbf{S}\|_F^2,$$

and so we require that if $\alpha$ is feasible, then $\alpha \geq \|(\mathbf{A}^{(i)}\mathbf{S})(\mathbf{VS})^\dagger\mathbf{R}^{(i)})((\mathbf{A}^{(i)}\mathbf{S})(\mathbf{VS})^\dagger\mathbf{R}^{(i)})^\dagger\mathbf{A}^{(i)}\mathbf{S} - \mathbf{A}^{(i)}\mathbf{S}\|_F^2$. Unfortunately, we do not know $\mathbf{V}$, so instead we use a polynomial solver to check whether there exists such a $\mathbf{V}$. We remark that similar guessing strategies were employed by Razenshteyn et al. (2016); Kumar et al. (2019); Ban et al. (2019); Velingker et al. (2023) and in particular, Razenshteyn et al. (2016) also uses a polynomial system in conjunction with the guessing strategy. Thus we write $\mathbf{Y} = \mathbf{VS}$ and its pseudoinverse $\mathbf{W} = (\mathbf{VS})^\dagger$ and check whether there exists a satisfying assignment to the above inequality, given the constraints (1) $\mathbf{YWY} = \mathbf{Y}$, (2) $\mathbf{WYW} = \mathbf{W}$, and (3) $\mathbf{A}^{(i)}\mathbf{SWR}^{(i)}$ has orthonormal columns. Note that since $\mathbf{V} \in \mathbb{R}^{k \times d}$, then implementing the polynomial solver naïvely could require $kd$ variables and thus use $2^{\Omega(dk)}$ runtime. Instead, we note that we only work with $\mathbf{VS}$, which has dimension $k \times m$ for $m = \mathcal{O}\left(\frac{k^2}{\varepsilon^2}\log \ell\right)$, so that the polynomial solver only uses $2^{\text{poly}(mk)}$ time.

We now show a crucial structural property that allows us to distinguish between the case where a guess $\alpha$ for the optimal value $\mathsf{OPT}$ exceeds $(1+\varepsilon)\mathsf{OPT}$ or is smaller than $(1-\varepsilon)\mathsf{OPT}$ by simply looking at a polynomial system solver on an affine embedding.

**Lemma 3.1.** *Let $\mathbf{V} \in \mathbb{R}^{k \times d}$ be the optimal solution to the fair low-rank approximation problem for inputs $\mathbf{A}^{(1)}, \ldots, \mathbf{A}^{(\ell)}$, where $\mathbf{A}^{(i)} \in \mathbb{R}^{n_i \times d}$, and suppose $\mathsf{OPT} = \max_{i \in [\ell]} \|\mathbf{A}^{(i)}\mathbf{V}^\dagger\mathbf{V} - \mathbf{A}^{(i)}\|_F^2$. Let $\mathbf{S}$ be an affine embedding for $\mathbf{V}$ and let $\mathbf{W} = (\mathbf{VS})^\dagger \in \mathbb{R}^{k \times m}$. For $i \in [\ell]$, let $\mathbf{Z}^{(i)} = \mathbf{A}^{(i)}\mathbf{SW} \in \mathbb{R}^{n_i \times k}$ and $\mathbf{R}^{(i)} \in \mathbb{R}^{k \times k}$ be defined so that $\mathbf{A}^{(i)}\mathbf{SWR}^{(i)}$ has orthonormal columns. If $\alpha \geq (1+\varepsilon)\cdot\mathsf{OPT}$, then for each $i \in [\ell]$, $\alpha \geq \|(\mathbf{A}^{(i)}\mathbf{SWR}^{(i)})(\mathbf{A}^{(i)}\mathbf{SWR}^{(i)})^\dagger\mathbf{A}^{(i)} - \mathbf{A}^{(i)}\|_F^2$. If $\alpha < (1-\varepsilon)\cdot\mathsf{OPT}$, then there exists $i \in [\ell]$, such that $\alpha < \|(\mathbf{A}^{(i)}\mathbf{SWR}^{(i)})(\mathbf{A}^{(i)}\mathbf{SWR}^{(i)})^\dagger\mathbf{A}^{(i)} - \mathbf{A}^{(i)}\|_F^2$.*

---

**Algorithm 3** Input to polynomial solver

**Input:** $\mathbf{A}^{(1)}, \ldots, \mathbf{A}^{(\ell)}, \mathbf{S}, \alpha$
**Output:** Feasibility of polynomial system
 1: **Polynomial variables**
 2: Let $\mathbf{Y} = (\mathbf{VS}) \in \mathbb{R}^{k \times m}$ be $mk$ variables
 3: Let $\mathbf{W} = (\mathbf{VS})^\dagger \in \mathbb{R}^{m \times k}$ be $mk$ variables
 4: Let $\mathbf{R}^{(i)} \in \mathbb{R}^{k \times k}$ for each $i \in [\ell]$ be $\ell k^2$ variables
 5: **System constraints**
 6: $\mathbf{YWY} = \mathbf{Y}$, $\mathbf{WYW} = \mathbf{W}$
 7: $\mathbf{A}^{(i)}\mathbf{SWR}^{(i)}$ has orthonormal columns
 8: $\alpha \geq \|(\mathbf{A}^{(i)}\mathbf{SWR}^{(i)})(\mathbf{A}^{(i)}\mathbf{SWR}^{(i)})^\dagger\mathbf{A}^{(i)} - \mathbf{A}^{(i)}\|_F^2$
 9: **Run polynomial system solver**
10: If feasible, output $\mathbf{V} = (\mathbf{A}^{(1)}\mathbf{SWR}^{(1)})^\dagger\mathbf{A}^{(1)}$. Otherwise, output $\perp$.

**Algorithm 4** $(1+\varepsilon)$-approximation for Fair Low-Rank Approximation

**Input:** $\mathbf{A}^{(i)} \in \mathbb{R}^{n_i \times d}$ for all $i \in [\ell]$, rank parameter $k > 0$, accuracy parameter $\varepsilon \in (0,1)$
**Output:** $(1+\varepsilon)$-approximation for fair low-rank approximation
 1: Let $\alpha$ be an $\ell$-approximation for the fair LRA problem
 2: Let $\mathbf{S}$ be generated from a random affine embedding distribution
 3: **while** Algorithm 3 on input $\mathbf{A}^{(1)}, \ldots, \mathbf{A}^{(\ell)}$, $\mathbf{S}$, and $\alpha$ does not return $\perp$ **do**
 4:     Let $\mathbf{V}$ be the output of Algorithm 3 on input $\mathbf{A}^{(1)}, \ldots, \mathbf{A}^{(\ell)}$, $\mathbf{S}$, and $\alpha$
 5:     $\alpha \leftarrow \frac{\alpha}{1+\varepsilon}$
 6: **return** $\mathbf{V}$

---

### 3.2 BICRITERIA ALGORITHM

To achieve polynomial time for our bicriteria algorithm, we can no longer use a polynomial system solver. Instead, we observe that for sufficiently large $p$, we have $\max \|\mathbf{x}\|_\infty = (1 \pm \varepsilon)\|\mathbf{x}\|_p$. Thus, in place of optimizing $\min_{\mathbf{V} \in \mathbb{R}^{k \times d}} \max_{i \in [\ell]} \|\mathbf{A}^{(i)}\mathbf{V}^\dagger\mathbf{V} - \mathbf{A}^{(i)}\|_F$, we instead optimize $\min_{\mathbf{V} \in \mathbb{R}^{k \times d}} \left(\sum_{i \in [\ell]} \|\mathbf{A}^{(i)}\mathbf{V}^\dagger\mathbf{V} - \mathbf{A}^{(i)}\|_F^p\right)^{1/p}$. However, the terms $\|\mathbf{A}^{(i)}\mathbf{V}^\dagger\mathbf{V} - \mathbf{A}^{(i)}\|_F^p$ are unwieldy to work with. Thus we instead use Dvoretzky's Theorem, i.e., Theorem 3.2, to embed $L_2$ into $L_p$, by generating matrices $\mathbf{G}$ and $\mathbf{H}$ so that $(1-\varepsilon)\|\mathbf{GMH}\|_p \leq \|\mathbf{M}\|_F \leq (1+\varepsilon)\|\mathbf{GMH}\|_p$, for all matrices $\mathbf{M} \in \mathbb{R}^{n \times d}$.

Now, writing $\mathbf{A} = \mathbf{A}^{(1)} \circ \ldots \circ \mathbf{A}^{(\ell)}$, it suffices to approximately solve $\min_{\mathbf{X} \in \mathbb{R}^{k \times d}} \|\mathbf{GAHSX} - \mathbf{GAH}\|_p$. Unfortunately, low-rank approximation with $L_p$ loss still cannot be approximated to $(1+\varepsilon)$-factor in polynomial time, and in fact $\mathbf{GAH}$ has dimension $n' \times d'$ with $n' \geq n$ and $d' \geq d$. Hence, we first apply dimensionality reduction by appealing to a result of Woodruff & Yasuda (2023a)

showing that there exists a matrix $\mathbf{S}$ that samples a "small" number of columns of $\mathbf{A}$ to provide a coarse bicriteria approximation to $L_p$ low-rank approximation. Now to lift the solution back to dimension $d$, we would like to solve regression problem $\min_{\mathbf{X}} \|\mathbf{GAHSX} - \mathbf{GAH}\|_p$.

To that end, we consider a Lewis weight sampling matrix $\mathbf{T}$ such that

$$\frac{1}{2}\|\mathbf{TGAHSX} - \mathbf{TGAH}\|_p \leq \|\mathbf{GAHSX} - \mathbf{GAH}\|_p \leq 2\|\mathbf{TGAHSX} - \mathbf{TGAH}\|_p.$$

We then note that $(\mathbf{TGAHS})^\dagger \mathbf{TGAH}$ is the minimizer of the problem $\min_{\mathbf{x}} \|\mathbf{TGAHSX} - \mathbf{TGAH}\|_F$, which only provides a small distortion to the $L_p$ regression problem, since $\mathbf{TGAH}$ has a small number of rows due to the dimensionality reduction. By Dvoretzky's Theorem, we have that $(\mathbf{TGAHS})^\dagger \mathbf{TGA}$ is a "good" approximate solution to the original fair low-rank approximation problem. The algorithm appears in full in Algorithm 5.

---

**Algorithm 5** Bicriteria approximation for fair low-rank approximation

---

**Input:** $\mathbf{A}^{(i)} \in \mathbb{R}^{n_i \times d}$ for all $i \in [\ell]$, rank parameter $k > 0$, trade-off parameter $c \in (0, 1)$
**Output:** Bicriteria approximation for fair low-rank approximation
1: Generate Gaussian matrices $\mathbf{G} \in \mathbb{R}^{n' \times n}, \mathbf{H} \in \mathbb{R}^{d \times d'}$ through Theorem 3.2
2: Let $\mathbf{S} \in \mathbb{R}^{n' \times t}, \mathbf{Z} \in \mathbb{R}^{t \times d'}$ be the output of Theorem 3.3 on input $\mathbf{GAH}$
3: Let $\mathbf{T} \in \mathbb{R}^{s \times n'}$ be a Lewis weight sampling matrix for $\mathbf{GAHSX} - \mathbf{GAH}$
4: Let $\widetilde{\mathbf{V}} \leftarrow (\mathbf{TGAHS})^\dagger (\mathbf{TGA})$
5: **return** $\widetilde{\mathbf{V}}$

---

We use the following notion of Dvoretzky's theorem to embed the problem into entrywise $L_p$ loss.

**Theorem 3.2** (Dvoretzky's Theorem, e.g., Theorem 1.2 in (Paouris et al., 2017)). *Let $p \geq 1$ be a parameter and let*

$$m \gtrsim m(n, p, \varepsilon) = \begin{cases} \frac{p^p n}{\varepsilon^2}, & \varepsilon \leq (Cp)^{p/2} n^{-\frac{p-2}{2(p-1)}} \\ \frac{(np)^{p/2}}{\varepsilon}, & \varepsilon \in \left( (Cp)^{p/2} n^{-\frac{p-2}{2(p-1)}}, \frac{1}{p} \right] \\ \frac{n^{p/2}}{p^{p/2} \varepsilon^{p/2}} \log^{p/2} \frac{1}{\varepsilon}, & \frac{1}{p} < \varepsilon < 1. \end{cases}$$

*Then there exists a family $\mathcal{G}$ of random scaled Gaussian matrices with dimension $\mathbb{R}^{m \times n}$ such that for $G \sim \mathcal{G}$, with probability at least $1 - \delta$, simultaneously for all $\mathbf{y} \in \mathbb{R}^n$, $(1 - \varepsilon)\|\mathbf{y}\|_2 \leq \|\mathbf{Gy}\|_p \leq (1 + \varepsilon)\|\mathbf{y}\|_2$.*

We use the following algorithm from (Woodruff & Yasuda, 2023a) to perform dimensionality reduction so that switching between $L_2$ and $L_p$ loss will incur smaller error. See also (Chierichetti et al., 2017).

**Theorem 3.3** (Theorem 1.5 in (Woodruff & Yasuda, 2023a)). *Let $\mathbf{A} \in \mathbb{R}^{n \times d}$ and let $k \geq 1$. Let $s = \mathcal{O}(k \log \log k)$. Then there exists a polynomial-time algorithm that outputs a matrix $\mathbf{S} \in \mathbb{R}^{d \times t}$ that samples $t = \mathcal{O}\left(k(\log \log k)(\log^2 d)\right)$ columns of $\mathbf{A}$ and a matrix $\mathbf{Z} \in \mathbb{R}^{t \times d}$ such that $\|\mathbf{A} - \mathbf{ASZ}\|_p \leq 2^p \cdot \mathcal{O}(\sqrt{s}) \cdot \min_{\mathbf{U} \in \mathbb{R}^{n \times k}, \mathbf{V} \in \mathbb{R}^{k \times d}} \|\mathbf{A} - \mathbf{UV}\|_p$.*

We recall the following construction to use Lewis weights to achieve an $L_p$ subspace embedding.

**Theorem 3.4** (Cohen & Peng (2015)). *Let $\varepsilon \in (0, 1)$ and $p \geq 2$. Let $\mathbf{A} \in \mathbb{R}^{n \times d}$ and $s = \mathcal{O}\left(d^{p/2} \log d\right)$. Then there exists a polynomial-time algorithm that outputs a matrix $\mathbf{S} \in \mathbb{R}^{s \times n}$ that samples and reweights $s$ rows of $\mathbf{A}$, such that with probability at least $0.99$, simultaneously for all $\mathbf{x} \in \mathbb{R}^d$, $(1 - \varepsilon)\|\mathbf{Ax}\|_p^p \leq \|\mathbf{SAx}\|_p^p \leq (1 + \varepsilon)\|\mathbf{Ax}\|_p^p$.*

We then show that Algorithm 5 provides a bicriteria approximation.

**Lemma 3.5.** *Let $\widetilde{\mathbf{V}}$ be the output of Algorithm 5. Then with probability at least $\frac{9}{10}$,*

$$\max_{i \in [\ell]} \|\mathbf{A}^{(i)} (\widetilde{\mathbf{V}})^\dagger \widetilde{\mathbf{V}} - \mathbf{A}^{(i)}\|_F \leq \ell^c \cdot 2^{1/c} \cdot \mathcal{O}\left(k(\log \log k)(\log d)\right) \max_{i \in [\ell]} \|\mathbf{A}^{(i)} (\widetilde{\mathbf{V}})^\dagger \widetilde{\mathbf{V}} - \mathbf{A}^{(i)}\|_F.$$

Since the generation of Gaussian matrices and the Lewis weight sampling matrix both only require polynomial time, it follows that our algorithm uses polynomial time overall. Hence, we have Theorem 1.6.

## 4  EMPIRICAL EVALUATIONS

In this section, we describe our empirical evaluations on socially fair regression.

**Law school dataset.** We used the Law School Admissions Councils National Longitudinal Bar Passage Study (Wightman, 1998), which has 22,407 observations. The task is to predict a student's first year GPA at law school via least squared regression, given the student's undergraduate GPA and whether or not they passed the bar exam, using race as the protected attribute. In particular, the dataset contains the following distribution of individuals self-identifying for each race: 980 "hisp", 1280 "black", 839 "asian", 17924 "white", 387 "other", and 997 providing no response.

**Experimental setup.** We compare Algorithm 1 and Algorithm 2. Intuitively, the former can be viewed as finding the optimal solution to the regression problem on the entire dataset, while the latter can be viewed as finding the optimal solution with respect to the socially fair regression objective. Thus the comparisons of the algorithms essentially measure the gain of fair algorithmic design for socially fair regression, i.e., how much the objective improves when using the optimal socially fair regression solution rather than the optimal regression solution to the entire dataset.

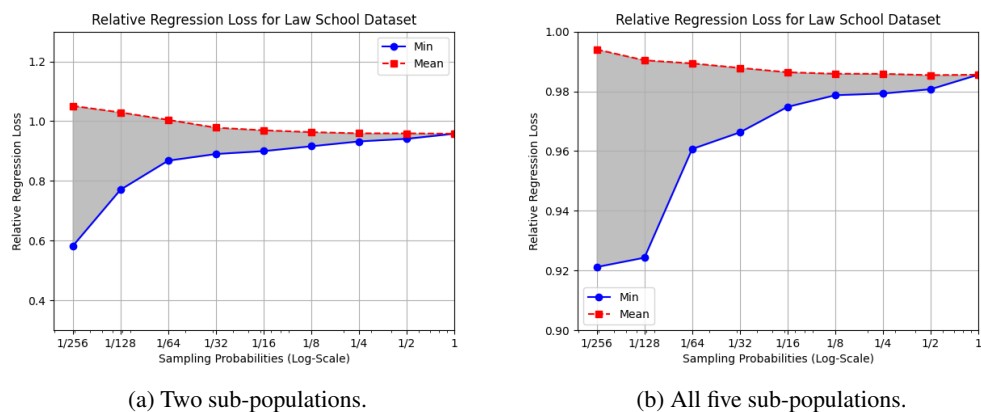

(a) Two sub-populations.  (b) All five sub-populations.

Fig. 1: Improvement by socially fair regression algorithm under linear least squares objective for the law school dataset when solution is computed using a subset of the data sampled at rate $p$, across 50 independent instances.

We run both algorithms over the entire law school dataset subsampling each group at rates $p \in \{2^{-i} | i \in \{0, \ldots, 9\}\}$. In other words, if group $i$ has $g_i$ observations in the entire dataset, then we randomly select a group of $\lfloor p \cdot g_i \rfloor$ observations to form the input to the algorithms. Using the parameter vector output by the algorithms, we then measure their corresponding cost to the socially fair regression objective on the entire dataset, i.e., the entire groups. We then compute the ratio of the outputs of the algorithms. That is, if Algorithm 1 outputs vector $\mathbf{u}$ and Algorithm 2 outputs vector $\mathbf{v}$, we compute $\frac{\max_{i \in [\ell]} \|\mathbf{A}^{(i)}\mathbf{v} - \mathbf{b}^{(i)}\|_2^2}{\max_{i \in [\ell]} \|\mathbf{A}^{(i)}\mathbf{u} - \mathbf{b}^{(i)}\|_2^2}$, across 50 independent instances for each value of $p$. We then plot the minimum and mean values of these instances for each value of $p$ in Figure 1, with Figure 1a denoting restrictions to two sub-populations and Figure 1b considering all five sub-populations.

**Results and discussion.** Our empirical evaluations show that our algorithms can perform significantly better for socially regression. In particular, for two sub-populations, the fair regression algorithm, Algorithm 2, can demonstrate more than $40\%$ improvement for the fair regression objective over the standard regression algorithm, Algorithm 1, i.e., Figure 1a with sampling rate $p = 1/256$. When considering all five sub-populations, the improvement can be as large as $8\%$, c.f., Figure 1b. We do notice that on average, the ratios of the performances for smaller values of $p$ can be larger than 1 for two sub-populations though this phenomenon does not present itself for all five sub-populations; we attribute this to large variances in the sampled observations resulting. On the other hand, for larger sampling rates, i.e., $p \geq 1/32$, the variance of the ratios of the performances becomes significantly smaller and the fair regression algorithm demonstrates a clear, albeit smaller, improvement over the standard regression algorithm for both two sub-populations and all five sub-populations. We present a number of additional empirical evaluations in Appendix E, including normalized loss across each group, synthetic datasets, larger numbers of features, and runtime analyses.

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

## A PRELIMINARIES

We use the notation $[n]$ to represent the set $\{1, \ldots, n\}$ for an integer $n \geq 1$. We use the notation $\mathrm{poly}(n)$ to represent a fixed polynomial in $n$ and we use the notation $\mathrm{polylog}(n)$ to represent $\mathrm{poly}(\log n)$. We use $\mathrm{poly}(n)$ to denote a fixed polynomial in $n$ and $\mathrm{polylog}(n)$ to denote $\mathrm{poly}(\log n)$. We say an event holds with high probability if it holds with probability $1 - \frac{1}{\mathrm{poly}(n)}$.

We generally use bold-font variables to represent vectors and matrices, whereas we use default-font variables to represent scalars. For a matrix $\mathbf{A} \in \mathbb{R}^{n \times d}$, we use $\mathbf{A}_i$ to represent the $i$-th row of $\mathbf{A}$ and $\mathbf{A}^{(j)}$ to represent the $j$-th column of $\mathbf{A}$. We use $A_{i,j}$ to represent the entry in the $i$-th row and $j$-th column of $\mathbf{A}$. For $p \geq 1$, we use

$$\|\mathbf{A}\|_p = \left( \sum_{i \in [n]} \sum_{j \in [d]} A_{i,j}^p \right)^{1/p}$$

to represent the entrywise $L_p$ norm of $\mathbf{A}$ and we use

$$\|\mathbf{A}\|_F = \left( \sum_{i \in [n]} \sum_{j \in [d]} A_{i,j}^2 \right)^{1/2}$$

to represent the Frobenius norm of $\mathbf{A}$, which is simply the entrywise $L_2$ norm of $\mathbf{A}$. We use define the $L_{p,q}$ of $\mathbf{A}$ as the $L_p$ norm of the vector consisting of the $L_q$ norms of each row of $\mathbf{A}$, so that

$$\|\mathbf{A}\|_{p,q} = \left( \sum_{i \in [n]} \left( \sum_{j \in [d]} (A_{i,j})^q \right)^{p/q} \right)^{1/p}.$$

Similarly, we use $\|\mathbf{A}\|_{(p,q)}$ to denote the $L_p$ norm of the vector consisting of the $L_q$ norms of each column of $\mathbf{A}$. Equivalently, we have $\|\mathbf{A}\|_{(p,q)} = \|\mathbf{A}^\top\|_{p,q}$, so that

$$\|\mathbf{A}\|_{(p,q)} = \left( \sum_{j \in [d]} \left( \sum_{i \in [n]} (A_{i,j})^q \right)^{p/q} \right)^{1/p}.$$

We use $\circ$ to represent vertical stacking of matrices, so that

$$\mathbf{A}^{(1)} \circ \ldots \circ \mathbf{A}^{(m)} = \begin{bmatrix} \mathbf{A}^{(1)} \\ \vdots \\ \mathbf{A}^{(m)} \end{bmatrix}.$$

## A.1 REGRESSION AND LOW-RANK APPROXIMATION

In this section, we briefly describe some common techniques used to handle both regression and low-rank approximation, thus presenting multiple unified approaches for both problems. Thus in light of the abundance of techniques that can be used to handle both problems, it is somewhat surprising that socially fair regression and socially fair low-rank approximation exhibit vastly different complexities.

**Closed form solutions.**    Given the regression problem $\min_{\mathbf{x} \in \mathbb{R}^d} \|\mathbf{A}\mathbf{x} - \mathbf{b}\|_2$ for an input matrix $\mathbf{A} \in \mathbb{R}^{n \times d}$ and a label vector $\mathbf{b} \in \mathbb{R}^n$, the closed form solution for the minimizer is $\mathbf{A}^\dagger \mathbf{b} = \text{argmin}_{\mathbf{x} \in \mathbb{R}^d} \|\mathbf{A}\mathbf{x} - \mathbf{b}\|_2$, where $\mathbf{A}^\dagger$ is the Moore-Penrose pseudoinverse of $\mathbf{A}$.

Similarly, given an input matrix $\mathbf{A}$ and a rank parameter $k > 0$, there exists a closed form solution for the minimizer $\text{argmin}_{\mathbf{V} \in \mathbb{R}^{k \times d}} \|\mathbf{A} - \mathbf{A}\mathbf{V}^\top \mathbf{V}\|_F^2$. Specifically, by the Eckart-Young-Mirsky theorem Eckart & Young (1936), the minimizer is the top $k$ right singular vectors of $\mathbf{A}$.

**Dimensionality reduction.**    We next recall a unified set of dimensionality reduction techniques for both linear regression and low-rank approximation. We consider the "sketch-and-solve" paradigm, so that for both problems, we first acquire a low-dimension representation of the problem, and find the optimal solution in the low dimension using the above closed-form solutions. For "good" designs of the low-dimension representations, the low-dimension solution will also be near-optimal for the original problem.

We first observe that oblivious linear sketches serve as a common dimensionality reduction for both linear regression and low-rank approximation. For example, it is known Woodruff (2014) that there exists a family of Gaussian random matrices $\mathcal{G}_1$ from which $\mathbf{S} \sim \mathcal{G}_1$ satisfies with high probability,

$$(1 - \varepsilon)\|\mathbf{S}\mathbf{A}\mathbf{x} - \mathbf{S}\mathbf{b}\|_2 \leq \|\mathbf{A}\mathbf{x} - \mathbf{b}\|_2 \leq (1 + \varepsilon)\|\mathbf{S}\mathbf{A}\mathbf{x} - \mathbf{S}\mathbf{b}\|_2,$$

simultaneously for all $\mathbf{x} \in \mathbb{R}^d$. Similarly, there exists Woodruff (2014) a family of Gaussian random matrices $\mathcal{G}_2$ from which $\mathbf{S} \sim \mathcal{G}_1$ satisfies with high probability, that the row space of $\mathbf{S}\mathbf{A}$ contains a $(1 + \varepsilon)$-approximation of the optimal low-rank approximation to $\mathbf{A}$.

Alternatively, we can achieve dimensionality reduction for both linear regression and low-rank approximation by sampling a small subset of the input in related ways for both problems. For linear regression, we can generate a random matrix $\mathbf{S}$ by sampling rows of $[\mathbf{A} \quad \mathbf{b}]$ by their leverage scores Drineas et al. (2006a;b); Magdon-Ismail (2010); Woodruff (2014). In this manner, we again achieve a matrix $\mathbf{S}$ such that with high probability,

$$(1 - \varepsilon)\|\mathbf{S}\mathbf{A}\mathbf{x} - \mathbf{S}\mathbf{b}\|_2 \leq \|\mathbf{A}\mathbf{x} - \mathbf{b}\|_2 \leq (1 + \varepsilon)\|\mathbf{S}\mathbf{A}\mathbf{x} - \mathbf{S}\mathbf{b}\|_2,$$

simultaneously for all $\mathbf{x} \in \mathbb{R}^d$. For low-rank approximation, we can generate a random matrix $\mathbf{S}$ by sampling rows of $\mathbf{A}$ with the related ridge-leverage scores Cohen et al. (2017). Then with high probability, we have for all $\mathbf{V} \in \mathbb{R}^{k \times d}$,

$$(1-\varepsilon)\|\mathbf{SA} - \mathbf{SAV}^\top\mathbf{V}\|_F^2 \leq \|\mathbf{A} - \mathbf{AV}^\top\mathbf{V}\|_F^2 \leq (1+\varepsilon)\|\mathbf{SA} - \mathbf{SAV}^\top\mathbf{V}\|_F^2.$$

# B  MISSING PROOFS FROM SECTION 2

**Theorem 2.1.** *The optimal solution to the standard regression problem that computes a vector $\widehat{\mathbf{x}} \in \mathbb{R}^d$ also satisfies $\max_{i\in[\ell]} \|\mathbf{A}^{(i)}\widehat{\mathbf{x}} - \mathbf{b}^{(i)}\|_2 \leq \ell \cdot \min_{\mathbf{x}\in\mathbb{R}^d} \max_{i\in[\ell]} \|\mathbf{A}^{(i)}\mathbf{x} - \mathbf{b}^{(i)}\|_2$, i.e., the algorithm outputs a $\ell$-approximation to the fair regression problem. For $L_2$ loss, the algorithm uses $\mathcal{O}\left(nd^{\omega-1}\right)$ runtime, where $n = n_1 + \ldots + n_\ell$ and $\omega$ is the matrix multiplication exponent.*

*Proof.* Let $\widehat{\mathbf{x}} = \operatorname{argmin}_{\mathbf{x}\in\mathbb{R}^d} \|\mathbf{Ax} - \mathbf{b}\|$, where $\mathbf{A} = \mathbf{A}^{(1)} \circ \ldots \circ \mathbf{A}^{(\ell)}$ and $\mathbf{b} = \mathbf{b}^{(1)} \circ \ldots \circ \mathbf{b}^{(\ell)}$. Let $\mathbf{x}^* = \operatorname{argmin}_{\mathbf{x}\in\mathbb{R}^d} \max_{i\in[\ell]} \|\mathbf{A}^{(i)}\mathbf{x} - \mathbf{b}^{(i)}\|$. Then by the optimality of $\widehat{\mathbf{x}}$, we have

$$\|\mathbf{A}\widehat{\mathbf{x}} - \mathbf{b}\| \leq \|\mathbf{Ax}^* - \mathbf{b}\|.$$

By triangle inequality,

$$\|\mathbf{Ax}^* - \mathbf{b}\|_2 \leq \sum_{i=1}^{\ell} \|\mathbf{A}^{(i)}\mathbf{x}^* - \mathbf{b}^{(i)}\| \leq \ell \cdot \max_{i\in[\ell]} \|\mathbf{A}^{(i)}\mathbf{x}^* - \mathbf{b}^{(i)}\|.$$

Therefore,

$$\|\mathbf{A}\widehat{\mathbf{x}} - \mathbf{b}\| \leq \ell \cdot \max_{i\in[\ell]} \|\mathbf{A}^{(i)}\mathbf{x}^* - \mathbf{b}^{(i)}\|,$$

so that $\widehat{\mathbf{x}}$ produces an $\ell$-approximation to the fair regression problem.

Finally, note that for $L_2$ loss, the closed form solution of $\widehat{\mathbf{x}}$ is $\widehat{\mathbf{x}} = (\mathbf{A}^{(i)})^\dagger \mathbf{b}^{(i)}$ and can computed in runtime $\mathcal{O}\left(nd^{\omega-1}\right)$. $\qquad\square$

## B.1  ALGORITHMS FOR $L_1$ AND $L_2$ REGRESSION

We now give an efficient algorithm for $(1+\varepsilon)$-approximation for fair $L_1$ regression using linear programs.

**Definition B.1** (Linear program formulation for min-max $L_1$ formulation). *Let $n = \sum_{i=1}^{\ell} n_i$. Given $\mathbf{A}^{(i)} \in \mathbb{R}^{n_i \times d}$, $\mathbf{b}^{(i)} \in \mathbb{R}^{n_i}$ and a parameter $L > 0$, the linear program formulation (with $d + n$ variables and $O(n)$ constraints) can be written as follows*

$$\min_{x\in\mathbb{R}^d, t\in\mathbb{R}^{\sum_{i=1}^{\ell} n_i}} \langle x, \mathbf{1}_d \rangle$$

$$\text{subject to } (\mathbf{A}^{(i)}x - \mathbf{b}^{(i)})_j \leq t_{i,j} \cdot \mathbf{1}_d, \qquad \forall i \in [\ell], \forall j \in [n_i]$$

$$(\mathbf{A}^{(i)}x - \mathbf{b}^{(i)})_j \geq -t_{i,j} \cdot \mathbf{1}_d, \qquad \forall i \in [\ell], \forall j \in [n_i]$$

$$t_{i,j} \geq 0, \qquad \forall i \in [\ell], \forall j \in [n_i]$$

$$\sum_{j=1}^{n_i} t_{i,j} \leq L, \qquad \forall i \in [\ell]$$

*Here $\mathbf{1}_d$ is a length-$d$ where all the entries are ones.*

**Observation B.2.** *The linear program in Definition B.1 has a feasible solution if and only if*

$$L \geq \min_{\mathbf{x}\in\mathbb{R}^d} \max_{i\in[\ell]} \|\mathbf{A}^{(i)}\mathbf{x} - \mathbf{b}^{(i)}\|_1.$$

**Theorem B.3.** *There exists an algorithm that uses $\mathcal{O}\left((n+d)^\omega\right)$ runtime and outputs whether the linear program in Definition B.1 has a feasible solution. Here $\omega \approx 2.373$.*

*Proof.* The proof directly follows using linear programming solver (Cohen et al., 2019; Jiang et al., 2021) as a black-box. $\qquad\square$

Given Theorem B.3 and the $\ell$-approximation algorithm as a starting point, we can achieve a $(1 + \varepsilon)$-approximation to fair $L_1$ regression using binary search in $\mathcal{O}\left(\frac{1}{\varepsilon} \log \ell\right)$ iterations.

**Definition B.4** (Quadratic program formulation for min-max $L_2$ formulation). *Given $\mathbf{A}^{(i)} \in \mathbb{R}^{n_i \times d}$, $\mathbf{b}^{(i)} \in \mathbb{R}^{n_i}$ and a parameter $L > 0$, the quadratic constraint quadratic program formulation can be written as follows*

$$\min_{x \in \mathbb{R}^d} x^\top I_d x$$
$$\text{subject to } x^\top (A^{(i)})^\top A^{(i)} x - 2\langle A^{(i)} x, b^{(i)} \rangle + \|b^{(i)}\|_2^2 \leq L, \qquad \forall i \in [\ell]$$

We now describe a similar approach for fair $L_2$ regression using a quadratic program.

**Observation B.5.** *The quadratically constrained quadratic program in Definition B.4 has a feasible solution if and only if*

$$L \geq \min_{\mathbf{x} \in \mathbb{R}^d} \max_{i \in [\ell]} \|\mathbf{A}^{(i)} \mathbf{x} - \mathbf{b}^{(i)}\|_2^2.$$

**Theorem B.6.** *There exists an algorithm that uses $\mathrm{poly}(n + d)$ runtime and outputs whether the quadratically constrained quadratic program in Definition B.4 has a feasible solution. Here $n = \sum_{i=1}^{\ell} n_i$. The polynomial factor is at least $4$.*

*Proof.* This directly follows using semi-definite programming solver as a black-box (Jiang et al., 2020a; Huang et al., 2022). Note that in general, quadratic programming is NP-hard to solve, however the formulation we have, all the matrices are semi-definite matrix. Thus, it's straightforward to reduce it to a SDP and then run SDP solver. $\square$

## B.2 ON CONVEX SOLVERS

We first require the following definition of a separation oracle.

**Definition B.7** (Separation oracle). *Given a set $K \subset \mathbb{R}^d$ and $\varepsilon > 0$, a separation oracle for $K$ is a function that takes an input $x \in \mathbb{R}^d$, either outputs that $x$ is in $K$ or outputs a separating hyperplane, i.e., a half-space of the form $H := \{z \mid c^\top z \leq c^\top x + b\} \supseteq K$ with $b \leq \varepsilon \|c\|_2$ and $c \neq 0^d$.*

We next recall the following statement on the runtime of convex solvers given access to a separation oracle.

**Theorem B.8** (Lee et al. (2015); Jiang et al. (2020b)). *Suppose there exists a set $K$ that is contained in a box of radius $R$ and a separation oracle that, given a point $x$ and using time $T$, either outputs that $x$ is in $K$ or outputs a separating hyperplane. Then there exists an algorithm that either finds a point in $K$ or proves that $K$ does not contain a ball of radius $\varepsilon$. The algorithm uses $\mathcal{O}(dT \log \kappa) + d^3 \cdot \mathrm{polylog}\, \kappa$ runtime, for $\kappa = \frac{dR}{\varepsilon}$.*

We use the following reduction from the computation of subgradients to separation oracles, given by Lemma 38 in (Lee et al., 2015).

**Lemma B.9** (Lee et al. (2015)). *Given $\alpha > 0$, suppose $K$ is a set defined by $\{x \in [-\Delta, \Delta]^d \mid \|Ax\| \leq \alpha\}$, where $\Delta = \mathrm{poly}(n)$ and the subgradient of $\|Ax\|$ can be computed in $\mathrm{poly}(n, d)$ time. Then there exists a separation oracle for $K$ that uses $\mathrm{poly}\left(\frac{nd}{\varepsilon}\right)$ time.*

Putting things together, we now have our main algorithm for fair regression:

**Theorem 1.1.** *Let $n_1, \ldots, n_\ell$ be positive integers and for each $i \in [\ell]$, let $\mathbf{A}^{(i)} \in \mathbb{R}^{n_i \times d}$ and $\mathbf{b}^{(i)} \in \mathbb{R}^{n_i}$. Let $\Delta = \mathrm{poly}(n)$ for $n = n_1 + \ldots + n_\ell$ and let $\varepsilon \in (0, 1)$. For any norm $\|\cdot\|$ whose subgradient can be computed in $\mathrm{poly}(n, d)$ time, there exists an algorithm that outputs $\mathbf{x}^* \in [-\Delta, \Delta]^d$ such that $\max_{i \in [\ell]} \|\mathbf{A}^{(i)} \mathbf{x}^* - \mathbf{b}^{(i)}\| \leq \varepsilon + \min_{\mathbf{x} \in [-\Delta, \Delta]^d} \max_{i \in [\ell]} \|\mathbf{A}^{(i)} \mathbf{x} - \mathbf{b}^{(i)}\|$. The algorithm uses $\mathrm{poly}\left(n, d, \frac{1}{\varepsilon}\right)$ runtime.*

*Proof.* Recall that every norm is convex, since $\|\lambda \mathbf{u} + (1 - \lambda)\mathbf{v}\| \leq \lambda \|\mathbf{u}\| + (1 - \lambda)\|\mathbf{v}\|$ by triangle inequality. Therefore, the function $g(\mathbf{x}) := \max_{i \in [\ell]} \{\|\mathbf{A}^{(i)} \mathbf{x} - \mathbf{b}^{(i)}\|\}$ is convex because the maximum of convex functions is also a convex function. Hence, the objective $\min_{\mathbf{x}} g(\mathbf{x})$ is the minimization of a convex function and can be solved using a convex program. $\square$

For further runtime improvements for sparse inputs, we recall the following notion of subspace embeddings.

**Definition B.10** (Subspace embedding)*. Given an input matrix $\mathbf{A} \in \mathbb{R}^{n \times d}$ and an accuracy parameter $\varepsilon \in (0, 1)$, a subspace embedding for $\mathbf{A}$ is a matrix $\mathbf{M} \in \mathbb{R}^{m \times d}$ such that for all $\mathbf{x} \in \mathbb{R}^d$, we have*

$$(1 - \varepsilon)\|\mathbf{A}\mathbf{x}\|_p \leq \|\mathbf{M}\mathbf{x}\|_p \leq (1 + \varepsilon)\|\mathbf{A}\mathbf{x}\|_p.$$

Constructions of subspace embeddings have been well-studied. We utilize the following constructions:

**Definition B.11** (Cohen & Peng (2015); Lee (2016); Jambulapati et al. (2022); Woodruff & Yasuda (2023b))*. Given an input matrix $\mathbf{A} \in \mathbb{R}^{n \times d}$ and an accuracy parameter $\varepsilon \in (0, 1)$, let $m = \mathcal{O}\left(\frac{d^{p/2}}{\varepsilon^2} \cdot \log^2 d \cdot \log n\right)$ for $p > 2$ and $m = \tilde{\mathcal{O}}\left(\frac{d}{\varepsilon^2}\right)$ for $p \leq 2$. There exists an algorithm outputs a matrix $\mathbf{M} \in \mathbb{R}^{m \times d}$ such that with probability $0.99$, simultaneously for all $\mathbf{x} \in \mathbb{R}^d$, we have*

$$(1 - \varepsilon)\|\mathbf{A}\mathbf{x}\|_p \leq \|\mathbf{M}\mathbf{x}\|_p \leq (1 + \varepsilon)\|\mathbf{A}\mathbf{x}\|_p.$$

*The algorithm uses $\tilde{\mathcal{O}}\left(\mathrm{nnz}(\mathbf{A})\right) + \mathrm{poly}\left(d, \frac{1}{\varepsilon}\right)$ runtime.*

# C  MISSING PROOFS FROM SECTION 3

## C.1  LOWER BOUND

We first show in Section C.1 that it is NP-hard to approximate fair low-rank approximation within any constant factor in polynomial time and moreover, under the exponential time hypothesis, it requires exponential time to achieve a constant factor approximation. We then give missing details from Section 3.1 and in Section 3.2.

Given points $\mathbf{v}^{(1)}, \ldots, \mathbf{v}^{(n)} \in \mathbb{R}^d$, their outer $(d - k)$-radius is defined as the minimum, over all $k$-dimensional linear subspaces, of the maximum Euclidean distance of these points to the subspace. We define this problem as $\mathsf{Subspace}(k, \infty)$. It is known that it is NP-hard to approximate the $\mathsf{Subspace}(n - 1, \infty)$ problem within any constant factor:

**Theorem C.1** (Brieden et al. (2000); Deshpande et al. (2011))*. The $\mathsf{Subspace}(n - 1, \infty)$ problem is NP-hard to approximate within any constant factor.*

Utilizing the NP-hardness of approximation of the $\mathsf{Subspace}(n - 1, \infty)$ problem, we show the NP-hardness of approximation of fair low-rank approximation.

**Theorem 1.3.** *Fair low-rank approximation is NP-hard to approximate within any constant factor.*

*Proof.* Given an instance $\mathbf{v}^{(1)}, \ldots, \mathbf{v}^{(n)} \in \mathbb{R}^d$ of $\mathsf{Subspace}(n - 1, \infty)$ with $n < d$, we set $\ell = k = n - 1$ and $\mathbf{A}^{(i)} = \mathbf{v}^{(i)}$ for all $i \in [n]$. Then for a $k$-dimensional linear subspace $\mathbf{V} \in \mathbb{R}^{k \times d}$, we have that $\|\mathbf{A}^{(i)}\mathbf{V}^\top\mathbf{V} - \mathbf{A}^{(i)}\|_F^2$ is the distance from $\mathbf{v}^{(i)}$ to the subspace. Hence, $\max_{i \in [\ell]} \|\mathbf{A}^{(i)}\mathbf{V}^\top\mathbf{V} - \mathbf{A}^{(i)}\|_F^2$ is the maximum Euclidean distance of these points to the subspace and so the fair low-rank approximation problem is exactly $\mathsf{Subspace}(n - 1, \infty)$. By Theorem C.1, the $\mathsf{Subspace}(n - 1, \infty)$ problem is NP-hard to approximate within any constant factor. Thus, fair low-rank approximation is NP-hard to approximate within any constant factor. $\square$

We next introduce a standard complexity assumption beyond NP-hardness. Recall that in the 3-SAT problem, the input is a Boolean satisfiability problem written in conjunctive normal form, consisting of $n$ clauses, each with 3 literals, either a variable or the negation of a variable. The goal is to determine whether there exists a Boolean assignment to the variables to satisfy the formula.

**Hypothesis C.2** (Exponential time hypothesis Impagliazzo & Paturi (2001))*. The 3-SAT problem requires $2^{\Omega(n)}$ runtime.*

Observe that while NP-hardness simply conjectures that the 3-SAT problem cannot be solved in polynomial time, the exponential time hypothesis conjectures that the 3-SAT problem requires *exponential* time.

We remark that in the context of Theorem C.1, Brieden et al. (2000) showed the hardness of approximation of $\mathsf{Subspace}(n - 1, \infty)$ through a reduction from the Max-Not-All-Equal-3-SAT problem, whose NP-hardness itself is shown through a reduction from 3-SAT. Thus under the exponential time hypothesis, Max-Not-All-Equal-3-SAT problem requires $2^{\Omega(n)}$ to solve. Then it follows that:

**Theorem C.3** (Brieden et al. (2000); Deshpande et al. (2011)). *Assuming the exponential time hypothesis, then the $\mathsf{Subspace}(n - 1, \infty)$ problem requires $2^{n^{\Omega(1)}}$ time to approximate within any constant factor.*

It follows that under the exponential time hypothesis, any constant-factor approximation to socially fair low-rank approximation requires exponential time.

**Theorem 1.4.** *Under the exponential time hypothesis, the fair low-rank approximation requires $2^{k^{\Omega(1)}}$ time to approximate within any constant factor.*

*Proof.* Given an instance $\mathbf{v}^{(1)}, \ldots, \mathbf{v}^{(k)} \in \mathbb{R}^d$ of $\mathsf{Subspace}(k - 1, \infty)$ with $k < d$, we set $\ell = k - 1$ and $\mathbf{A}^{(i)} = \mathbf{v}^{(i)}$ for all $i \in [k]$. Then for a $(k - 1)$-dimensional linear subspace $\mathbf{V} \in \mathbb{R}^{(k-1) \times d}$, we have that $\|\mathbf{A}^{(i)} \mathbf{V}^\top \mathbf{V} - \mathbf{A}^{(i)}\|_F^2$ is the distance from $\mathbf{v}^{(i)}$ to the subspace. Hence, $\max_{i \in [\ell]} \|\mathbf{A}^{(i)} \mathbf{V}^\top \mathbf{V} - \mathbf{A}^{(i)}\|_F^2$ is the maximum Euclidean distance of these points to the subspace and so the fair low-rank approximation problem is exactly $\mathsf{Subspace}(k - 1, \infty)$. By Theorem C.3, the $\mathsf{Subspace}(k - 1, \infty)$ problem requires $2^{k^{\Omega(1)}}$ time to approximate within any constant factor. Thus, fair low-rank approximation requires $2^{k^{\Omega(1)}}$ time to approximate within any constant factor. □

### C.2 MISSING PROOFS FROM SECTION 3.1

We first recall the following result for polynomial system satisfiability solvers.

**Theorem C.4** (Renegar (1992a;b); Basu et al. (1996)). *Given a polynomial system $P(x_1, \ldots, x_n)$ over real numbers and $m$ polynomial constraints $f_i(x_1, \ldots, x_n) \otimes_i 0$, where $\otimes \in \{>, \geq, =, \neq, \leq, <\}$ for all $i \in [m]$, let $d$ denote the maximum degree of all the polynomial constraints and let $B$ denote the maximum size of the bit representation of the coefficients of all the polynomial constraints. Then there exists an algorithm that determines whether there exists a solution to the polynomial system $P$ in time $(md)^{\mathcal{O}(n)} \cdot \mathrm{poly}(B)$.*

To apply Theorem C.4, we utilize the following statement upper bounding the sizes of the bit representation of the coefficients of the polynomial constraints in our system.

**Theorem C.5** (Jeronimo et al. (2013)). *Let $\mathcal{T} = \{x \in \mathbb{R}^n \mid f_1(x) \geq 0, \ldots, f_m(x) \geq 0\}$ be defined by $m$ polynomials $f_i(x_1, \ldots, x_n)$ for $i \in [m]$ with degrees bounded by an even integer $d$ and coefficients of magnitude at most $M$. Let $\mathcal{C}$ be a compact connected component of $\mathcal{T}$. Let $g(x_1, \ldots, x_n)$ be a polynomial of degree at most $d$ with integer coefficients of magnitude at most $M$. Then the minimum nonzero magnitude that $g$ takes over $\mathcal{C}$ is at least $(2^{4-n/2} \widetilde{M} d^n)^{-n 2^n d^n}$, where $\widetilde{M} = \max(M, 2n + 2m)$.*

To perform dimensionality reduction, recall the following definition of affine embedding.

**Definition C.6** (Affine embedding). *We say a matrix $\mathbf{S} \in \mathbb{R}^{n \times m}$ is an affine embedding for a matrix $\mathbf{A} \in \mathbb{R}^{d \times n}$ and a vector $\mathbf{b} \in \mathbb{R}^n$ if we have*

$$(1 - \varepsilon) \|\mathbf{x A} - \mathbf{b}\|_F^2 \leq \|\mathbf{x A S} - \mathbf{b S}\|_F^2 \leq (1 + \varepsilon) \|\mathbf{x A} - \mathbf{b}\|_F^2,$$

*for all vectors $\mathbf{x} \in \mathbb{R}^d$.*

We then apply the following affine embedding construction.

**Lemma C.7** (Lemma 11 in (Cohen et al., 2015)). *Given $\delta, \varepsilon \in (0, 1)$ and a rank parameter $k > 0$, let $m = \mathcal{O}\left(\frac{k^2}{\varepsilon^2} \log \frac{1}{\delta}\right)$. For any matrix $\mathbf{A} \in \mathbb{R}^{d \times n}$, there exists a family $\mathcal{S}$ of random matrices in*

$\mathbb{R}^{n \times m}$, *such that for* $\mathbf{S} \sim \mathcal{S}$, *we have that with probability at least* $1 - \delta$, $\mathbf{S}$ *is a one-sided affine embedding for a matrix* $\mathbf{A} \in \mathbb{R}^{d \times n}$ *and a vector* $\mathbf{b} \in \mathbb{R}^n$.

We now show a crucial structural property that allows us to distinguish between the case where a guess $\alpha$ for the optimal value OPT exceeds $(1 + \varepsilon)$OPT or is smaller than $(1 - \varepsilon)$OPT by simply looking at a polynomial system solver on an affine embedding.

**Lemma 3.1.** *Let* $\mathbf{V} \in \mathbb{R}^{k \times d}$ *be the optimal solution to the fair low-rank approximation problem for inputs* $\mathbf{A}^{(1)}, \ldots, \mathbf{A}^{(\ell)}$, *where* $\mathbf{A}^{(i)} \in \mathbb{R}^{n_i \times d}$, *and suppose* $\mathsf{OPT} = \max_{i \in [\ell]} \|\mathbf{A}^{(i)} \mathbf{V}^\dagger \mathbf{V} - \mathbf{A}^{(i)}\|_F^2$. *Let* $\mathbf{S}$ *be an affine embedding for* $\mathbf{V}$ *and let* $\mathbf{W} = (\mathbf{VS})^\dagger \in \mathbb{R}^{k \times m}$. *For* $i \in [\ell]$, *let* $\mathbf{Z}^{(i)} = \mathbf{A}^{(i)} \mathbf{SW} \in \mathbb{R}^{n_i \times k}$ *and* $\mathbf{R}^{(i)} \in \mathbb{R}^{k \times k}$ *be defined so that* $\mathbf{A}^{(i)} \mathbf{SWR}^{(i)}$ *has orthonormal columns. If* $\alpha \geq (1 + \varepsilon) \cdot \mathsf{OPT}$, *then for each* $i \in [\ell]$, $\alpha \geq \|(\mathbf{A}^{(i)} \mathbf{SWR}^{(i)})(\mathbf{A}^{(i)} \mathbf{SWR}^{(i)})^\dagger \mathbf{A}^{(i)} - \mathbf{A}^{(i)}\|_F^2$. *If* $\alpha < (1 - \varepsilon) \cdot \mathsf{OPT}$, *then there exists* $i \in [\ell]$, *such that* $\alpha < \|(\mathbf{A}^{(i)} \mathbf{SWR}^{(i)})(\mathbf{A}^{(i)} \mathbf{SWR}^{(i)})^\dagger \mathbf{A}^{(i)} - \mathbf{A}^{(i)}\|_F^2$.

*Proof.* By the Pythagorean theorem, we have that

$$(\mathbf{A}^{(i)} \mathbf{S})(\mathbf{VS})^\dagger = \operatorname*{argmin}_{\mathbf{X}^{(i)}} \|\mathbf{X}^{(i)} \mathbf{VS} - \mathbf{A}^{(i)} \mathbf{S}\|_F^2.$$

Thus, $\mathbf{Z}^{(i)} = \operatorname{argmin}_{\mathbf{X}^{(i)}} \|\mathbf{X}^{(i)} \mathbf{VS} - \mathbf{A}^{(i)} \mathbf{S}\|_F^2$ for $\mathbf{Z}^{(i)} = \mathbf{A}^{(i)} \mathbf{SW} \in \mathbb{R}^{n_i \times k}$ and $\mathbf{W} = (\mathbf{VS})^\dagger \in \mathbb{R}^{m \times k}$.

Let $\mathbf{R}^{(i)} \in \mathbb{R}^{k \times k}$ be defined so that $\mathbf{A}^{(i)} \mathbf{SWR}^{(i)}$ has orthonormal columns. Thus, we have

$$\begin{aligned}
\|(\mathbf{Z}^{(i)} \mathbf{R}^{(i)})(\mathbf{Z}^{(i)} \mathbf{R}^{(i)})^\dagger \mathbf{A}^{(i)} \mathbf{S} - \mathbf{A}^{(i)} \mathbf{S}\|_F^2 &= \|(\mathbf{Z}^{(i)})(\mathbf{Z}^{(i)})^\dagger \mathbf{A}^{(i)} \mathbf{S} - \mathbf{A}^{(i)} \mathbf{S}\|_F^2 \\
&= \|\mathbf{Z}^{(i)} \mathbf{VS} - \mathbf{A}^{(i)} \mathbf{S}\|_F^2 \\
&= \min_{\mathbf{X}^{(i)}} \|\mathbf{X}^{(i)} \mathbf{VS} - \mathbf{A}^{(i)} \mathbf{S}\|_F^2,
\end{aligned}$$

by the definition of $\mathbf{Z}^{(i)}$.

Suppose $\alpha \geq (1 + \varepsilon) \cdot \mathsf{OPT}$. Since $\mathbf{S}$ is an affine embedding for $\mathbf{V}$, then we have that for all $\mathbf{X}^{(i)} \in \mathbb{R}^{n_i \times k}$,

$$\|\mathbf{X}^{(i)} \mathbf{VS} - \mathbf{A}^{(i)} \mathbf{S}\|_F^2 \leq (1 + \varepsilon) \|\mathbf{X}^{(i)} \mathbf{V} - \mathbf{A}^{(i)}\|_F^2.$$

In particular, we have

$$\min_{\mathbf{X}^{(i)}} \|\mathbf{X}^{(i)} \mathbf{VS} - \mathbf{A}^{(i)} \mathbf{S}\|_F^2 \leq (1 + \varepsilon) \min_{\mathbf{X}^{(i)}} \|\mathbf{X}^{(i)} \mathbf{V} - \mathbf{A}^{(i)}\|_F^2 \leq (1 + \varepsilon)\mathsf{OPT}.$$

Then from the above argument, we have

$$\begin{aligned}
\|(\mathbf{Z}^{(i)} \mathbf{R}^{(i)})(\mathbf{Z}^{(i)} \mathbf{R}^{(i)})^\dagger \mathbf{A}^{(i)} \mathbf{S} - \mathbf{A}^{(i)} \mathbf{S}\|_F^2 &= \min_{\mathbf{X}^{(i)}} \|\mathbf{X}^{(i)} \mathbf{VS} - \mathbf{A}^{(i)} \mathbf{S}\|_F^2 \\
&\leq (1 + \varepsilon)\mathsf{OPT} \leq \alpha.
\end{aligned}$$

Since $\mathbf{Z}^{(i)} = \mathbf{A}^{(i)} \mathbf{SW} \in \mathbb{R}^{n_i \times k}$, then for $\alpha \geq (1 + \varepsilon) \cdot \mathsf{OPT}$, we have that for each $i \in [\ell]$,

$$\alpha \geq \|(\mathbf{A}^{(i)} \mathbf{SWR}^{(i)})(\mathbf{A}^{(i)} \mathbf{SWR}^{(i)})^\dagger \mathbf{A}^{(i)} - \mathbf{A}^{(i)}\|_F^2.$$

On the other hand, suppose $\alpha < (1 - \varepsilon) \cdot \mathsf{OPT}$. Let $i \in [\ell]$ be fixed so that

$$\mathsf{OPT} = \min_{\mathbf{X}^{(i)}} \|\mathbf{X}^{(i)} \mathbf{V} - \mathbf{A}^{(i)}\|_F^2.$$

Since $\mathbf{S}$ is an affine embedding for $\mathbf{V}$, we have that for all $\mathbf{X}^{(i)} \in \mathbb{R}^{n_i \times k}$,

$$(1 - \varepsilon)\|\mathbf{X}^{(i)} \mathbf{V} - \mathbf{A}^{(i)}\|_F^2 \leq \|\mathbf{X}^{(i)} \mathbf{VS} - \mathbf{A}^{(i)} \mathbf{S}\|_F^2,$$

Therefore,

$$(1 - \varepsilon) \min_{\mathbf{X}^{(i)}} \|\mathbf{X}^{(i)} \mathbf{V} - \mathbf{A}^{(i)}\|_F^2 \leq \min_{\mathbf{X}^{(i)}} \|\mathbf{X}^{(i)} \mathbf{VS} - \mathbf{A}^{(i)} \mathbf{S}\|_F^2$$

From the above, we have

$$\|(\mathbf{Z}^{(i)} \mathbf{R}^{(i)})(\mathbf{Z}^{(i)} \mathbf{R}^{(i)})^\dagger \mathbf{A}^{(i)} \mathbf{S} - \mathbf{A}^{(i)} \mathbf{S}\|_F^2 = \min_{\mathbf{X}^{(i)}} \|\mathbf{X}^{(i)} \mathbf{VS} - \mathbf{A}^{(i)} \mathbf{S}\|_F^2.$$

Hence, putting these relations together,

$$\alpha < (1 - \varepsilon)\mathsf{OPT} = (1 - \varepsilon)\min_{\mathbf{X}^{(i)}}\|\mathbf{X}^{(i)}\mathbf{V} - \mathbf{A}^{(i)}\|_F^2$$
$$\leq \min_{\mathbf{X}^{(i)}}\|\mathbf{X}^{(i)}\mathbf{V}\mathbf{S} - \mathbf{A}^{(i)}\mathbf{S}\|_F^2$$
$$= \|(\mathbf{Z}^{(i)}\mathbf{R}^{(i)})(\mathbf{Z}^{(i)}\mathbf{R}^{(i)})^{\dagger}\mathbf{A}^{(i)}\mathbf{S} - \mathbf{A}^{(i)}\mathbf{S}\|_F^2,$$

as desired. $\qquad\square$

We can thus utilize the structural property of Lemma 3.1 by using the polynomial system solver in Algorithm 3 on an affine embedding.

**Corollary C.8.** *If $\alpha \geq (1 + \varepsilon) \cdot \mathsf{OPT}$, then Algorithm 3 outputs a vector $\mathbf{U} \in \mathbb{R}^{n \times k}$ such that*

$$\alpha \geq \|\mathbf{U}\mathbf{U}^{\dagger}\mathbf{A}^{(i)} - \mathbf{A}^{(i)}\|_F^2.$$

*If $\alpha < (1 - \varepsilon) \cdot \mathsf{OPT}$, then Algorithm 3 outputs $\perp$*

Correctness of Algorithm 4 then follows from Corollary C.8 and binary search on $\alpha$.

We now analyze the runtime of Algorithm 4.

**Lemma C.9.** *The runtime of Algorithm 4 is at most $\frac{1}{\varepsilon}\operatorname{poly}(n) \cdot (2\ell)^{\mathcal{O}(N)}$, for $n = \sum_{i=1}^{\ell} n_i$ and $N = \operatorname{poly}\left(\ell, k, \frac{1}{\varepsilon}\right)$.*

*Proof.* Suppose the coefficients of $\mathbf{A}^{(i)}$ are bounded in magnitude by $2^{\operatorname{poly}(n)}$, where $n = \sum_{i=1}^{\ell} n_i$. The number of variables in the polynomial system is at most

$$N := 2mk + \ell k^2 = \operatorname{poly}\left(\ell, k, \frac{1}{\varepsilon}\right).$$

Each of the $\mathcal{O}(\ell)$ polynomial constraints has degree at most 20. Thus by Theorem C.5, the minimum nonzero magnitude that any polynomial constraint takes over $\mathcal{C}$ is at least $(2^{4-N/2}2^{\operatorname{poly}(n)}20^N)^{-N2^N20^N}$. Hence, the maximum bit representation required is $B = \operatorname{poly}(n) \cdot 2^{\mathcal{O}(N)}$. By Theorem C.4, the runtime of the polynomial system solver is at most $(\mathcal{O}(\ell) \cdot 20)^{\mathcal{O}(N)} \cdot \operatorname{poly}(B) = \operatorname{poly}(n) \cdot (2\ell)^{\mathcal{O}(N)}$. We require at most $\mathcal{O}\left(\frac{1}{\varepsilon}\log\ell\right)$ iterations of the polynomial system solver. Thus, the total runtime is at most $\frac{1}{\varepsilon}\operatorname{poly}(n) \cdot (2\ell)^{\mathcal{O}(N)}$. $\qquad\square$

Putting things together, we have:

**Theorem 1.5.** *Given an accuracy parameter $\varepsilon \in (0, 1)$, there exists an algorithm that outputs $\widetilde{\mathbf{V}} \in \mathbb{R}^{k \times d}$ such that with probability at least $\frac{2}{3}$, $\max_{i \in [\ell]}\|\mathbf{A}^{(i)}(\widetilde{\mathbf{V}})^{\dagger}\widetilde{\mathbf{V}} - \mathbf{A}^{(i)}\|_F \leq (1 + \varepsilon) \cdot \min_{\mathbf{V} \in \mathbb{R}^{k \times d}}\max_{i \in [\ell]}\|\mathbf{A}^{(i)}\mathbf{V}^{\dagger}\mathbf{V} - \mathbf{A}^{(i)}\|_F$. The algorithm uses runtime $\frac{1}{\varepsilon}\operatorname{poly}(n) \cdot (2\ell)^{\mathcal{O}(N)}$, for $n = \sum_{i=1}^{\ell} n_i$ and $N = \operatorname{poly}\left(\ell, k, \frac{1}{\varepsilon}\right)$.*

### C.3 Missing Proofs from Section 3.2

**Lemma C.10.** *Let $\varepsilon \in (0, 1)$ and $\mathbf{x} \in \mathbb{R}^{\ell}$ and let $p = \mathcal{O}\left(\frac{1}{\varepsilon}\log\ell\right)$. Then $\|\mathbf{x}\|_{\infty} \leq \|\mathbf{x}\|_p \leq (1 + \varepsilon)\|\mathbf{x}\|_{\infty}$.*

*Proof.* Since it is clear that $\|\mathbf{x}\|_{\infty} \leq \|\mathbf{x}\|_p$, then it remains to prove $\|\mathbf{x}\|_p \leq (1 + \varepsilon)\|\mathbf{x}\|_{\infty}$. Note that we have $\|\mathbf{x}\|_p^p \leq \|\mathbf{x}\|_{\infty}^p \cdot \ell$. To achieve $\ell^{1/p} \leq (1 + \varepsilon)$, it suffices to have $\frac{1}{p}\log\ell \leq \log(1 + \varepsilon)$. Since $\log(1 + \varepsilon) = \mathcal{O}(\varepsilon)$ for $\varepsilon \in (0, 1)$, then for $p = \mathcal{O}\left(\frac{1}{\varepsilon}\log\ell\right)$, we have $\ell^{1/p} \leq (1 + \varepsilon)$, and the desired claim follows. $\qquad\square$

**Lemma 3.5.** *Let $\widetilde{\mathbf{V}}$ be the output of Algorithm 5. Then with probability at least $\frac{9}{10}$,*

$$\max_{i \in [\ell]}\|\mathbf{A}^{(i)}(\widetilde{\mathbf{V}})^{\dagger}\widetilde{\mathbf{V}} - \mathbf{A}^{(i)}\|_F \leq \ell^c \cdot 2^{1/c} \cdot \mathcal{O}\left(k(\log\log k)(\log d)\right)\max_{i \in [\ell]}\|\mathbf{A}^{(i)}(\widetilde{\mathbf{V}})^{\dagger}\widetilde{\mathbf{V}} - \mathbf{A}^{(i)}\|_F.$$

*Proof.* We have

$$\max_{i \in [\ell]} \|\mathbf{A}^{(i)}(\widetilde{\mathbf{V}})^\dagger \widetilde{\mathbf{V}} - \mathbf{A}^{(i)}\|_F \leq \left( \sum_{i \in [\ell]} \|\mathbf{A}^{(i)}(\widetilde{\mathbf{V}})^\dagger \widetilde{\mathbf{V}} - \mathbf{A}^{(i)}\|_F^p \right)^{1/p}.$$

Let $\mathbf{A} = \mathbf{A}^{(1)} \circ \ldots \circ \mathbf{A}^{(\ell)}$ and let

$$\widetilde{\mathbf{U}} := \mathbf{AHS}.$$

For $i \in [\ell]$, let $\widetilde{\mathbf{U}^{(i)}}$ be the matrix of $\widetilde{\mathbf{U}}$ whose rows correspond with the rows of $\mathbf{A}^{(i)}$ in $\mathbf{A}$, i.e., let $\widetilde{\mathbf{U}^{(i)}}$ be the $i$-th block of rows of $\widetilde{\mathbf{U}}$.

By the optimality of $\mathbf{A}^{(i)}(\widetilde{\mathbf{V}})^\dagger$ with respect to the Frobenius norm, we have

$$\left( \sum_{i \in [\ell]} \|\mathbf{A}^{(i)}(\widetilde{\mathbf{V}})^\dagger \widetilde{\mathbf{V}} - \mathbf{A}^{(i)}\|_F^p \right)^{1/p} \leq \left( \sum_{i \in [\ell]} \|\widetilde{\mathbf{U}^{(i)}} \widetilde{\mathbf{V}} - \mathbf{A}^{(i)}\|_F^p \right)^{1/p}$$

By Dvoretzky's Theorem, Theorem 3.2, with distortion $\varepsilon = \Theta(1)$, we have that with probability at least 0.99,

$$\left( \sum_{i \in [\ell]} \|\widetilde{\mathbf{U}^{(i)}} \widetilde{\mathbf{V}} - \mathbf{A}^{(i)}\|_F^p \right)^{1/p} \leq 2 \left( \sum_{i \in [\ell]} \|\mathbf{G} \widetilde{\mathbf{U}^{(i)}} \widetilde{\mathbf{V}} \mathbf{H} - \mathbf{GA}^{(i)} \mathbf{H}\|_p^p \right)^{1/p},$$

where we use $\| \cdot \|_p$ to denote the entry-wise $p$-norm. Writing $\widetilde{\mathbf{U}} = \widetilde{\mathbf{U}^{(1)}} \circ \ldots \circ \widetilde{\mathbf{U}^{(\ell)}}$, then we have

$$\left( \sum_{i \in [\ell]} \|\mathbf{G} \widetilde{\mathbf{U}^{(i)}} \widetilde{\mathbf{V}} \mathbf{H} - \mathbf{GA}^{(i)} \mathbf{H}\|_p^p \right)^{1/p} = \|\mathbf{G} \widetilde{\mathbf{U}} \widetilde{\mathbf{V}} \mathbf{H} - \mathbf{GAH}\|_p.$$

By the choice of the Lewis weight sampling matrix $\mathbf{T}$, we have that with probability 0.99,

$$\|\mathbf{G} \widetilde{\mathbf{U}} \widetilde{\mathbf{V}} \mathbf{H} - \mathbf{GAH}\|_p \leq 2\|\mathbf{TG} \widetilde{\mathbf{U}} \widetilde{\mathbf{V}} \mathbf{H} - \mathbf{TGAH}\|_p$$
$$\leq 2\|\mathbf{TG} \widetilde{\mathbf{U}} \widetilde{\mathbf{V}} \mathbf{H} - \mathbf{TGAH}\|_{(p,2)}$$
$$= 2\|\mathbf{TGAHS}(\mathbf{TGAHS})^\dagger (\mathbf{TGA})\mathbf{H} - \mathbf{TGAH}\|_{(p,2)}.$$

Here $\|\mathbf{M}\|_{(p,2)}$ denotes the $L_p$ norm of the vector consisting of the $L_2$ norms of the columns of $\mathbf{M}$. By optimality of $(\mathbf{TGAHS})^\dagger (\mathbf{TGA})\mathbf{H}$ for the choice of $\mathbf{X}$ in the minimization problem

$$\min_{\mathbf{X} \in \mathbb{R}^{t \times d'}} \|\mathbf{TGAHSX} - \mathbf{TGAH}\|_{(p,2)},$$

we have

$$\|\mathbf{TGAHS}(\mathbf{TGAHS})^\dagger (\mathbf{TGA})\mathbf{H} - \mathbf{TGAH}\|_{(p,2)} = \min_{\mathbf{X} \in \mathbb{R}^{t \times d'}} \|\mathbf{TGAHSX} - \mathbf{TGAH}\|_{(p,2)}.$$

Since $\mathbf{S} \in \mathbb{R}^{n' \times t}$ and $\mathbf{T}$ is a Lewis weight sampling matrix for $\mathbf{GAHSX} - \mathbf{GAH}$, then $\mathbf{T}$ has $t$ rows, where $t = \mathcal{O}\left( k(\log \log k)(\log^2 d) \right)$ by Theorem 3.3. Thus, each column of $\mathbf{TGAH}$ has $t$ entries, so that

$$\min_{\mathbf{X} \in \mathbb{R}^{t \times d'}} \|\mathbf{TGAHSX} - \mathbf{TGAH}\|_{(p,2)} \leq \sqrt{t} \min_{\mathbf{X} \in \mathbb{R}^{t \times d'}} \|\mathbf{TGAHSX} - \mathbf{TGAH}\|_p.$$

By the choice of the Lewis weight sampling matrix $\mathbf{T}$, we have that with probability 0.99,

$$\min_{\mathbf{X} \in \mathbb{R}^{t \times d'}} \|\mathbf{TGAHSX} - \mathbf{TGAH}\|_p \leq 2 \min_{\mathbf{X} \in \mathbb{R}^{t \times d'}} \|\mathbf{GAHSX} - \mathbf{GAH}\|_p.$$

By Theorem 3.3, we have that with probability 0.99,

$$\min_{\mathbf{X} \in \mathbb{R}^{t \times d'}} \|\mathbf{GAHSX} - \mathbf{GAH}\|_p \leq 2^p \cdot \mathcal{O}\left( \sqrt{s} \right) \cdot \min_{\mathbf{U} \in \mathbb{R}^{n' \times t}, \mathbf{V} \in \mathbb{R}^{t \times d'}} \|\mathbf{UV} - \mathbf{GAH}\|_p,$$

for $s = \mathcal{O}\left(k \log \log k\right)$. Let $\mathbf{V}^* = \operatorname{argmin}_{\mathbf{V} \in \mathbb{R}^{k \times d}} \max_{i \in [\ell]} \|\mathbf{A}^{(i)} - \mathbf{A}^{(i)} \mathbf{V}^\dagger \mathbf{V}\|_p$. Then since $\mathbf{UV}$ has rank $t$ with $t \geq k$, we have

$$\min_{\mathbf{U} \in \mathbb{R}^{n' \times t}, \mathbf{V} \in \mathbb{R}^{t \times d'}} \|\mathbf{UV} - \mathbf{GAH}\|_p \leq \|\mathbf{GA}(\mathbf{V}^*)^\dagger \mathbf{V}^* \mathbf{H} - \mathbf{GAH}\|_p$$

$$= \left( \sum_{i \in [\ell]} \|\mathbf{GA}^{(i)}(\mathbf{V}^*)^\dagger \mathbf{V}^* \mathbf{H} - \mathbf{GA}^{(i)} \mathbf{H}\|_p^p \right)^{1/p}.$$

By Dvoretzky's Theorem, Theorem 3.2, with distortion $\varepsilon = \Theta(1)$, we have that with probability at least $0.99$,

$$\left( \sum_{i \in [\ell]} \|\mathbf{GA}^{(i)}(\mathbf{V}^*)^\dagger \mathbf{V}^* \mathbf{H} - \mathbf{GA}^{(i)} \mathbf{H}\|_p^p \right)^{1/p} \leq 2 \left( \sum_{i \in [\ell]} \|\mathbf{A}^{(i)}(\mathbf{V}^*)^\dagger \mathbf{V}^* - \mathbf{A}^{(i)}\|_F^p \right)^{1/p}.$$

For $p = c \log \ell$ with $c < 1$, we have

$$\left( \sum_{i \in [\ell]} \|\mathbf{A}^{(i)}(\mathbf{V}^*)^\dagger \mathbf{V}^* - \mathbf{A}^{(i)}\|_F^p \right)^{1/p} \leq 2^{1/c} \max_{i \in [\ell]} \|\mathbf{A}^{(i)}(\widetilde{\mathbf{V}})^\dagger \widetilde{\mathbf{V}} - \mathbf{A}^{(i)}\|_F.$$

Putting together these inequalities successively, we ultimately have

$$\max_{i \in [\ell]} \|\mathbf{A}^{(i)}(\widetilde{\mathbf{V}})^\dagger \widetilde{\mathbf{V}} - \mathbf{A}^{(i)}\|_F \leq 2^p \cdot 2^{1/c} \cdot \mathcal{O}\left(\sqrt{st}\right) \max_{i \in [\ell]} \|\mathbf{A}^{(i)}(\widetilde{\mathbf{V}})^\dagger \widetilde{\mathbf{V}} - \mathbf{A}^{(i)}\|_F,$$

for $p = c \log \ell$, $s = \mathcal{O}\left(k \log \log k\right)$, and $t = \mathcal{O}\left(k (\log \log k)(\log^2 d)\right)$. Therefore, we have

$$\max_{i \in [\ell]} \|\mathbf{A}^{(i)}(\widetilde{\mathbf{V}})^\dagger \widetilde{\mathbf{V}} - \mathbf{A}^{(i)}\|_F \leq \ell^c \cdot 2^{1/c} \cdot \mathcal{O}\left(k (\log \log k)(\log d)\right) \max_{i \in [\ell]} \|\mathbf{A}^{(i)}(\widetilde{\mathbf{V}})^\dagger \widetilde{\mathbf{V}} - \mathbf{A}^{(i)}\|_F.$$

$\square$

## D    SOCIALLY FAIR COLUMN SUBSET SELECTION

In this section, we consider socially fair column subset selection, where the goal is to select a matrix $\mathbf{C} \in \mathbb{R}^{d \times k}$ that selects $k$ columns to minimize

$$\min_{\mathbf{C} \in \mathbb{R}^{d \times k}, \|\mathbf{C}\|_0 \leq k, \mathbf{B}^{(i)}} \max_{i \in [\ell]} \|\mathbf{B}^{(i)} \mathbf{C} - \mathbf{A}^i\|_F.$$

---

**Algorithm 6** Bicriteria approximation for fair column subset selection

---

**Input:** $\mathbf{A}^{(i)} \in \mathbb{R}^{n_i \times d}$ for all $i \in [\ell]$, rank parameter $k > 0$, trade-off parameter $c \in (0, 1)$
**Output:** Bicriteria approximation for fair column subset selection
  1: Acquire $\widetilde{\mathbf{V}}$ from Algorithm 5
  2: Generate Gaussian matrices $\mathbf{G} \in \mathbb{R}^{n' \times n}$ through Theorem 3.2
  3: Let $\mathbf{S} \in \mathbb{R}^{d \times k'}$ be a leverage score sampling matrix that samples $k' = \mathcal{O}\left(k \log k\right)$ columns of $\widetilde{\mathbf{V}}$
  4: $\mathbf{M}^{(i)} = \mathbf{S}^\dagger (\widetilde{\mathbf{V}})^\dagger \widetilde{\mathbf{V}}$ for all $i \in [\ell]$
  5: **return** $\mathbf{A}^{(i)} \mathbf{S}, \{\mathbf{M}^{(i)}\}$                    ▷Set of selected columns is given by $\mathbf{S}$

---

We first provide preliminaries on leverage score sampling.

**Definition D.1.** *Given a matrix $\mathbf{M} \in \mathbb{R}^{n \times d}$, we define the leverage score $\sigma_i$ of each row $\mathbf{m}_i$ with $i \in [n]$ by $\mathbf{m}_i(\mathbf{M}^\top \mathbf{M})^{-1} \mathbf{m}_i^\top$. Equivalently, for the singular value decomposition $\mathbf{M} = \mathbf{U} \boldsymbol{\Sigma} \mathbf{V}$, the leverage score of row $\mathbf{m}_i$ is also the squared row norm of $\mathbf{u}_i$.*

It is known that the sum of the leverage scores of the rows of a matrix can be bounded by the rank of the matrix.

**Theorem D.2** (Generalization of Foster's Theorem, Foster (1953)). *Given a matrix $\mathbf{M} \in \mathbb{R}^{n \times d}$, the sum of its leverage scores is* $\mathrm{rank}(\mathbf{M})$.

By sampling rows proportional to their leverage scores, we can obtain a subspace embedding as follows:

**Theorem D.3** (Leverage score sampling). *Drineas et al. (2006a;b); Magdon-Ismail (2010); Woodruff (2014) Given a matrix $\mathbf{M} \in \mathbb{R}^{n \times d}$, let $\sigma_i$ be the leverage score of the $i$-th row of $\mathbf{M}$. Suppose $p_i = \min(1, \sigma_i \log n)$ for each $i \in [n]$ and let $\mathbf{S}$ be a random diagonal matrix so that the $i$-th diagonal entry of $\mathbf{S}$ is $\frac{1}{\sqrt{p_i}}$ with probability $p_i$ and $0$ with probability $1 - p_i$. Then for all vectors $\mathbf{v} \in \mathbb{R}^n$,*

$$\mathbb{E}\left[\|\mathbf{S}\mathbf{v}\|_2^2\right] = \|\mathbf{v}\|_2^2$$

*and with probability at least $0.99$, for all vectors $\mathbf{x} \in \mathbb{R}^d$*

$$\frac{99}{100}\|\mathbf{M}\mathbf{x}\|_2 \leq \|\mathbf{S}\mathbf{M}\mathbf{x}\|_2 \leq \frac{101}{100}\|\mathbf{M}\mathbf{x}\|_2.$$

*Moreover, $\mathbf{S}$ has at most $\mathcal{O}(d \log n)$ nonzero entries with high probability.*

Since Theorem D.2 upper bounds the sum of the leverage scores by $d$ for an input matrix $\mathbf{M} \in \mathbb{R}^{n \times d}$, then Theorem D.3 shows that given the leverage scores of $\mathbf{M}$, it suffices to sample only $\mathcal{O}(d \log n)$ rows of $\mathbf{M}$ to achieve a constant factor subspace embedding of $\mathbf{M}$. Because the leverage scores of $\mathbf{M}$ can be computed directly from the singular value decomposition of $\mathbf{M}$, which can be computed in $\mathcal{O}(nd^\omega + dn^\omega)$ time where $\omega$ is the exponent of matrix multiplication, then the leverage scores of $\mathbf{M}$ can be computed in polynomial time.

Finally, we recall that to provide a constant factor approximation to $L_p$ regression, it suffices to compute a constant factor subspace embedding, e.g., through leverage score sampling. The proof is through the triangle inequality and is well-known among the active sampling literature (Chen & Price, 2019; Parulekar et al., 2021; Musco et al., 2022; Meyer et al., 2022; 2023), e.g., a generalization of Lemma 2.1 in Meyer et al. (2022). For completeness, we provide the proof below.

**Lemma D.4.** *Given a matrix $\mathbf{M} \in \mathbb{R}^{n \times d}$, let $\mathbf{S}$ be a matrix such that for all $\mathbf{x} \in \mathbb{R}^d$ and $\mathbf{v} \in \mathbb{R}^n$,*

$$\frac{11}{12}\|\mathbf{M}\mathbf{x}\|_2 \leq \|\mathbf{S}\mathbf{M}\mathbf{x}\|_2 \leq \frac{13}{12}\|\mathbf{M}\mathbf{x}\|_2, \qquad \mathbb{E}\left[\|\mathbf{S}\mathbf{v}\|_2^2\right] = \|\mathbf{v}\|_2^2.$$

*For a fixed $\mathbf{B} \in \mathbb{R}^{n \times m}$ where $\mathbf{B} = \mathbf{b}_1 \circ \ldots \circ \mathbf{b}_m$ with $\mathbf{b}_i \in \mathbb{R}^n$ for $i \in [m]$, let $\widetilde{\mathbf{x}}_i = (\mathbf{S}\mathbf{M})^\dagger(\mathbf{S}\mathbf{b}_i)$. Let $\widetilde{\mathbf{X}} = \widetilde{\mathbf{x}_1} \circ \ldots \circ \widetilde{\mathbf{x}_m}$. Then with probability at least $0.97$,*

$$\|\mathbf{M}\widetilde{\mathbf{X}} - \mathbf{B}\|_2 \leq 99 \min_{\mathbf{X}} \|\mathbf{M}\mathbf{X} - \mathbf{B}\|_2.$$

*Proof.* Let $\mathbf{X}^* = \mathrm{argmin}_{\mathbf{X}} \|\mathbf{M}\mathbf{X} - \mathbf{B}\|_2$ and $\mathsf{OPT} = \|\mathbf{M}\mathbf{X}^* - \mathbf{B}\|_2$. By triangle inequality,

$$\|\mathbf{S}\mathbf{M}\mathbf{X} - \mathbf{S}\mathbf{B}\|_2 \geq \|\mathbf{S}\mathbf{M}(\mathbf{X} - \mathbf{X}^*)\|_2 - \|\mathbf{S}\mathbf{M}\mathbf{X}^* - \mathbf{S}\mathbf{B}\|_2.$$

We have

$$\frac{99}{100}\|\mathbf{M}\mathbf{X}\|_2 \leq \|\mathbf{S}\mathbf{M}\mathbf{X}\|_2 \leq \frac{101}{100}\|\mathbf{M}\mathbf{X}\|_2$$

for all $\mathbf{X} \in \mathbb{R}^{n \times m}$. Thus,

$$\|\mathbf{S}\mathbf{M}\mathbf{X} - \mathbf{S}\mathbf{B}\|_2 \geq \frac{99}{100}\|\mathbf{M}(\mathbf{X} - \mathbf{X}^*)\|_2 - \|\mathbf{S}\mathbf{M}\mathbf{X}^* - \mathbf{S}\mathbf{B}\|_2.$$

By triangle inequality,

$$\|\mathbf{S}\mathbf{M}\mathbf{X} - \mathbf{S}\mathbf{B}\|_2 \geq \frac{99}{100}\left(\|\mathbf{M}\mathbf{X} - \mathbf{B}\|_2 - \|\mathbf{M}\mathbf{X}^* - \mathbf{B}\|_2\right) - \|\mathbf{S}\mathbf{M}\mathbf{X}^* - \mathbf{S}\mathbf{B}\|_2.$$

Since $\mathbb{E}\left[\|\mathbf{S}\mathbf{v}\|_2^2\right] = \|\mathbf{v}\|_2^2$ for all $\mathbf{x} \in \mathbb{R}^d$, then by concavity and Markov's inequality, we have that

$$\mathbf{Pr}\left[\|\mathbf{S}\mathbf{M}\mathbf{X}^* - \mathbf{S}\mathbf{B}\|_2 \geq 49\|\mathbf{M}\mathbf{X}^* - \mathbf{B}\|_2\right] \leq \frac{1}{49}.$$

Thus with probability at least 0.97,

$$\|\mathbf{SMX} - \mathbf{SB}\|_2 \geq \frac{99}{100} \left( \|\mathbf{MX} - \mathbf{B}\|_2 - \|\mathbf{MX}^* - \mathbf{B}\|_2 \right) - 49\|\mathbf{MX}^* - \mathbf{B}\|_2.$$

Now since we have $\|\mathbf{SMX}^* - \mathbf{SB}\|_2 \leq 49\|\mathbf{MX}^* - \mathbf{B}\|_2$ and $\|\mathbf{SM}\widetilde{\mathbf{X}} - \mathbf{SB}\|_2 \leq \|\mathbf{SMX}^* - \mathbf{SB}\|_2$, then

$$49\|\mathbf{MX}^* - \mathbf{B}\|_2 \geq \frac{99}{100} \left( \|\mathbf{M}\widetilde{\mathbf{X}} - \mathbf{B}\|_2 - \|\mathbf{MX}^* - \mathbf{B}\|_2 \right) - 49\|\mathbf{MX}^* - \mathbf{B}\|_2,$$

so that

$$\|\mathbf{M}\widetilde{\mathbf{X}} - \mathbf{B}\|_2 \leq 99\|\mathbf{MX}^* - \mathbf{B}\|_2,$$

as desired. □

We now give the correctness guarantees of Algorithm 6.

**Lemma D.5.** *Let* $\mathbf{S}, \mathbf{M}^{(1)}, \ldots, \mathbf{M}^{(\ell)}$ *be the output of Algorithm 6. Then with probability at least* $0.8$,

$$\max_{i \in [\ell]} \|\mathbf{A}^{(i)}\mathbf{SM}^{(i)} - \mathbf{A}^{(i)}\|_F \leq \ell^c \cdot 2^{1/c} \cdot \mathcal{O}\left(k(\log\log k)(\log d)\right) \min_{\mathbf{V} \in \mathbb{R}^{k \times d}} \max_{i \in [\ell]} \|\mathbf{A}^{(i)}\mathbf{V}^\dagger \mathbf{V} - \mathbf{A}^{(i)}\|_F.$$

*Proof.* Let $\widetilde{\mathbf{V}} \in \mathbb{R}^{t \times d}$ be the output of Algorithm 5, where $t = \mathcal{O}\left(k(\log\log k)(\log^2 d)\right)$. Then with probability at least $\frac{2}{3}$, we have

$$\max_{i \in [\ell]} \|\mathbf{A}^{(i)}(\widetilde{\mathbf{V}})^\dagger \widetilde{\mathbf{V}} - \mathbf{A}^{(i)}\|_F \leq \ell^c \cdot 2^{1/c} \cdot \mathcal{O}\left(k(\log\log k)(\log d)\right) \min_{\mathbf{V} \in \mathbb{R}^{k \times d}} \max_{i \in [\ell]} \|\mathbf{A}^{(i)}\mathbf{V}^\dagger \mathbf{V} - \mathbf{A}^{(i)}\|_F.$$

Therefore, we have

$$\max_{i \in [\ell]} \min_{\mathbf{B}^{(i)}} \|\mathbf{B}^{(i)}\widetilde{\mathbf{V}} - \mathbf{A}^{(i)}\|_F \leq \ell^c \cdot 2^{1/c} \cdot \mathcal{O}\left(k(\log\log k)(\log d)\right) \min_{\mathbf{V} \in \mathbb{R}^{k \times d}} \max_{i \in [\ell]} \|\mathbf{A}^{(i)}\mathbf{V}^\dagger \mathbf{V} - \mathbf{A}^{(i)}\|_F.$$

Let $p$ be a sufficiently large parameter to be fixed. By Dvoretzky's theorem, i.e., Theorem 3.2, with $\varepsilon = \mathcal{O}(1)$, we have

$$\max_{i \in [\ell]} \min_{\mathbf{B}^{(i)}} \|\mathbf{G}^{(i)}\mathbf{B}^{(i)}\widetilde{\mathbf{V}} - \mathbf{G}^{(i)}\mathbf{A}^{(i)}\|_{p,2} \leq \mathcal{O}(1) \cdot \max_{i \in [\ell]} \min_{\mathbf{B}^{(i)}} \|\mathbf{B}^{(i)}\widetilde{\mathbf{V}} - \mathbf{A}^{(i)}\|_F.$$

For sufficiently large $p = \mathcal{O}(\log \ell)$, we have by Lemma C.10,

$$\max_{i \in [\ell]} \min_{\mathbf{B}^{(i)}} \|\mathbf{B}^{(i)}\widetilde{\mathbf{V}} - \mathbf{A}^{(i)}\|_F. \leq \mathcal{O}(1) \cdot \min_{\mathbf{B}^{(i)}} \|\mathbf{G}^{(i)}\mathbf{B}^{(i)}\widetilde{\mathbf{V}} - \mathbf{G}^{(i)}\mathbf{A}^{(i)}\|_{p,2}.$$

By a change of variables, we have

$$\min_{\mathbf{X}} \|\mathbf{X}\widetilde{\mathbf{V}} - \mathbf{GA}\|_{p,2} \leq \min_{\mathbf{B}} \|\mathbf{GB}\widetilde{\mathbf{V}} - \mathbf{GA}\|_{p,2}.$$

Note that minimizing $\|\mathbf{X}\widetilde{\mathbf{V}} - \mathbf{GA}\|_{p,2}$ over all $\mathbf{X}$ corresponds to minimizing $\|\mathbf{X}_i\widetilde{\mathbf{V}} - \mathbf{G}_i\mathbf{A}\|_2$ over all $i \in [n']$. However, an arbitrary choice of $\mathbf{X}_i$ may not correspond to selecting columns of $\mathbf{A}$. Thus we apply a leverage score sampling matrix $\mathbf{S}$ to sample columns of $\widetilde{\mathbf{V}}$ which will correspondingly sample columns of $\mathbf{GA}$. Then by Lemma D.4, we have

$$\min_{\mathbf{X}} \|\mathbf{X}\widetilde{\mathbf{V}}\mathbf{S} - \mathbf{GAS}\|_{p,2} \leq \mathcal{O}(1) \cdot \min_{\mathbf{X}} \|\mathbf{X}\widetilde{\mathbf{V}} - \mathbf{GA}\|_{p,2}.$$

Putting these together, we have

$$\min_{\mathbf{X}} \|\mathbf{X}\widetilde{\mathbf{V}}\mathbf{S} - \mathbf{GAS}\|_{p,2} \leq \ell^c \cdot 2^{1/c} \cdot \mathcal{O}\left(k(\log\log k)(\log d)\right) \min_{\mathbf{V} \in \mathbb{R}^{k \times d}} \max_{i \in [\ell]} \|\mathbf{A}^{(i)}\mathbf{V}^\dagger \mathbf{V} - \mathbf{A}^{(i)}\|_F. \quad (1)$$

Let $S$ be the selected columns from Algorithm 6 by $\mathbf{S}$ and note that $\mathbf{A}^{(i)}\mathbf{S}\mathbf{S}^\dagger(\widetilde{\mathbf{V}})^\dagger\widetilde{\mathbf{V}}$ is in the column span of $\mathbf{A}^{(i)}\mathbf{S}$ for each $i \in [\ell]$. By Dvoretzky's theorem, i.e., Theorem 3.2, with $\varepsilon = \mathcal{O}(1)$ and a fixed parameter $p$, we have

$$\max_{i \in [\ell]} \|\mathbf{A}^{(i)}\mathbf{S}\mathbf{S}^\dagger(\widetilde{\mathbf{V}})^\dagger\widetilde{\mathbf{V}} - \mathbf{A}^{(i)}\|_F \leq \mathcal{O}(1) \cdot \max_{i \in [\ell]} \|\mathbf{G}^{(i)}\mathbf{A}^{(i)}\mathbf{S}\mathbf{S}^\dagger(\widetilde{\mathbf{V}})^\dagger\widetilde{\mathbf{V}} - \mathbf{G}^{(i)}\mathbf{A}^{(i)}\|_{p,2}.$$

For sufficiently large $p$, we have

$$\max_{i\in[\ell]}\|\mathbf{G}^{(i)}\mathbf{A}^{(i)}\mathbf{SS}^\dagger(\widetilde{\mathbf{V}})^\dagger\widetilde{\mathbf{V}} - \mathbf{G}^{(i)}\mathbf{A}^{(i)}\|_{p,2} \leq \mathcal{O}(1)\cdot\|\mathbf{GASS}^\dagger(\widetilde{\mathbf{V}})^\dagger\widetilde{\mathbf{V}} - \mathbf{GA}\|_{p,2}.$$

By the correctness of the leverage score sampling matrix, we have

$$\|\mathbf{GASS}^\dagger(\widetilde{\mathbf{V}})^\dagger\widetilde{\mathbf{V}} - \mathbf{GA}\|_{p,2} \leq \mathcal{O}(1)\cdot\|\mathbf{GASS}^\dagger(\widetilde{\mathbf{V}})^\dagger\widetilde{\mathbf{V}}\mathbf{S} - \mathbf{GAS}\|_{p,2}.$$

Observe that minimizing $\|\mathbf{X}\widetilde{\mathbf{V}}\mathbf{S} - \mathbf{GAS}\|_{p,2}$ over all $\mathbf{X}$ corresponds to minimizing $\|\mathbf{X}_i\widetilde{\mathbf{V}}\mathbf{S} - \mathbf{G}_i\mathbf{AS}\|_2$ over all $i\in[n']$. Moreover, the closed-form solution of the $L_2$ minization problem is

$$\mathbf{G}_i\mathbf{ASS}^\dagger(\widetilde{\mathbf{V}})^\dagger = \operatorname*{argmin}_{\mathbf{X}_i}\|\mathbf{X}_i\widetilde{\mathbf{V}}\mathbf{S} - \mathbf{G}_i\mathbf{AS}\|_2.$$

Therefore, we have

$$\|\mathbf{GASS}^\dagger(\widetilde{\mathbf{V}})^\dagger\widetilde{\mathbf{V}}\mathbf{S} - \mathbf{GAS}\|_{p,2} \leq \min_{\mathbf{X}}\|\mathbf{X}\widetilde{\mathbf{V}}\mathbf{S} - \mathbf{GAS}\|_{p,2}.$$

Putting things together, we have

$$\max_{i\in[\ell]}\|\mathbf{A}^{(i)}\mathbf{SS}^\dagger(\widetilde{\mathbf{V}})^\dagger\widetilde{\mathbf{V}} - \mathbf{A}^{(i)}\|_F \leq \mathcal{O}(1)\cdot\min_{\mathbf{X}}\|\mathbf{X}\widetilde{\mathbf{V}}\mathbf{S} - \mathbf{GAS}\|_{p,2}. \tag{2}$$

By Equation 1 and Equation 2 and a rescaling of the constant hidden inside the big Oh notation, we have

$$\max_{i\in[\ell]}\|\mathbf{A}^{(i)}\mathbf{SS}^\dagger(\widetilde{\mathbf{V}})^\dagger\widetilde{\mathbf{V}} - \mathbf{A}^{(i)}\|_F \leq \ell^c\cdot 2^{1/c}\cdot\mathcal{O}\left(k(\log\log k)(\log d)\right)\min_{\mathbf{V}\in\mathbb{R}^{k\times d}}\max_{i\in[\ell]}\|\mathbf{A}^{(i)}\mathbf{V}^\dagger\mathbf{V} - \mathbf{A}^{(i)}\|_F.$$

The desired claim then follows from the setting of $\mathbf{M}^{(i)} = \mathbf{S}^\dagger(\widetilde{\mathbf{V}})^\dagger\widetilde{\mathbf{V}}$ for $i\in[\ell]$ by Algorithm 6. $\qquad\square$

**Lemma D.6.** *The runtime of Algorithm 6 is polynomial in $n$ and $d$.*

**Theorem 1.7.** *Given input matrices $\mathbf{A}^{(i)} \in \mathbb{R}^{n_i\times d}$ with $n = \sum n_i$, there exists an algorithm that selects a set $S$ of $k' = \mathcal{O}(k\log k)$ columns such that with probability at least $\frac{2}{3}$, $S$ is a $\mathcal{O}(k(\log\log k)(\log d))$-approximation to the fair column subset selection problem. The algorithm uses runtime polynomial in $n$ and $d$.*

# E  ADDITIONAL EMPIRICAL EVALUATIONS

In this section, we present a number of additional results from our empirical evaluations.

## E.1  SOCIALLY FAIR REGRESSION

**Normalized group loss for law school dataset.**    A natural question to ask is whether the results may change for the law school dataset when the socially fair regression objective is normalized across each group. That is, instead of the objective function $\min_{\mathbf{x}\in\mathbb{R}^d}\max_{i\in[\ell]}\|\mathbf{A}^{(i)}\mathbf{x} - \mathbf{b}^{(i)}\|_2^2$ that minimizes the loss in each group, we consider the objective function $\min_{\mathbf{x}\in\mathbb{R}^d}\max_{i\in[\ell]}\frac{1}{n_i}\|\mathbf{A}^{(i)}\mathbf{x} - \mathbf{b}^{(i)}\|_2^2$ that minimizes the *average* loss in each group, where $n_i$ is the number of observations for group $i$. We show that our fair regression algorithm similarly (if not even more) performs better than the the standard regression algorithm on the normalized socially fair regression objective. Thus our experiments in Figure 2 show that for the law school dataset, considerations beyond the standard regression algorithms may be a worthy investment under the socially fair regression objective.

**Synthetic dataset.**    Finally, we perform experiments on a simple synthetic dataset, in particular to consider larger numbers of features. In this setup, we repeatedly generate matrices $\mathbf{A}^{(1)}, \mathbf{A}^{(2)}$ of size $10\times 3$, $40\times 10$, and $200\times 50$, with integer entries from 1 through $k$ for $k\in\{2,3,4,5,6,7,8,9,10\}$. We similarly generate vectors $\mathbf{b}^{(1)}, \mathbf{b}^{(2)}$ with the appropriate dimension and integer entries from 1 through $k$. We then compare the ratio of the socially fair objective values associated with the outputs of the fair regression algorithm and the standard regression algorithms. Our experimental results in Figure 3 similarly demonstrate the improvement of the fair regression algorithm over the standard regression algorithm across various values of $k$ and various matrix sizes. Finally, we remark that across all of our experiments, the fair regression algorithm used similar but always more runtime than the standard regression algorithm. We summarize these results in Table 1.

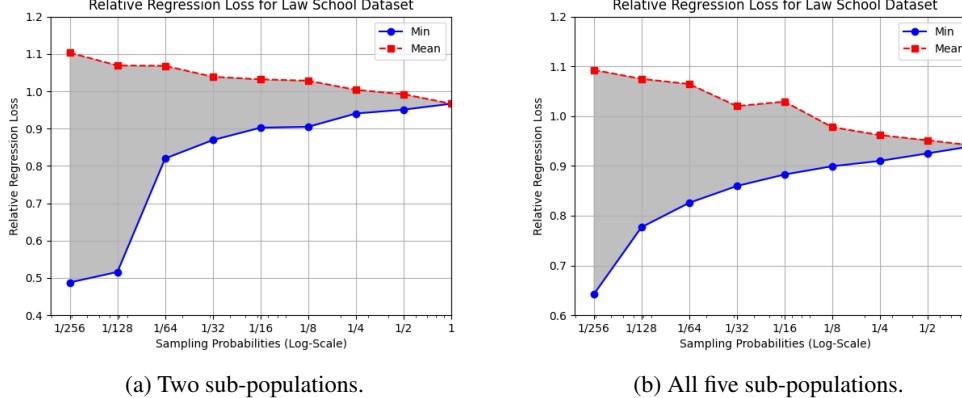

(a) Two sub-populations.

(b) All five sub-populations.

Fig. 2: Improvement by socially fair regression algorithm under linear least squares objective, ***normalized across each group*** for the law school dataset when solution is computed using a subset of the data sampled at rate $p$, across $50$ independent instances.

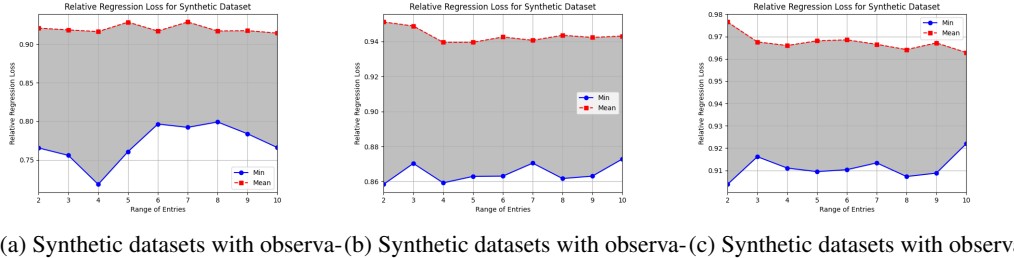

(a) Synthetic datasets with observations of dimension $10 \times 3$.

(b) Synthetic datasets with observations of dimension $40 \times 10$.

(c) Synthetic datasets with observations of dimension $200 \times 50$.

Fig. 3: Improvement by socially fair regression algorithm under linear least squares objective for synthetic dataset when dataset is generated with integer values from 1 through $k$, across $100$ independent instances.

| Dataset | Fair Algorithm | Standard Algorithm |
|---|---|---|
| Law school, two groups | 7.64 | 4.98 |
| Law school, five groups | 36.39 | 20.95 |
| Synthetic, $10 \times 3$ | 4.78 | 2.99 |
| Synthetic, $40 \times 10$ | 5.62 | 3.57 |
| Synthetic, $200 \times 50$ | 25.05 | 19.93 |

Table 1: Average runtime (in milliseconds) for algorithms across each dataset

## E.2  SOCIALLY FAIR LOW-RANK APPROXIMATION

Finally, we give a toy example using a synthetic dataset showing the importance of considering fairness in low-rank approximation.

**Synthetic dataset.**   We show that even for a simple dataset with four groups, each with a single observation across two features, the performance of the fair low-rank approximation algorithm can be much better than standard low-rank approximation algorithm on the socially fair low-rank objective. In this setup, we repeatedly generate matrices $\mathbf{A}^{(1)}, \mathbf{A}^{(2)}, \mathbf{A}^{(3)}, \mathbf{A}^{(4)} \in \{0,1\}^2$, with $\mathbf{A}^{(1)} = (1,0)$ and $\mathbf{A}^{(2)} = \mathbf{A}^{(3)} = \mathbf{A}^{(4)} = (0,1)$. The optimal fair low-rank solution is $\left(\frac{\sqrt{2}}{2}, \frac{\sqrt{2}}{2}\right)$ but due to the extra instances of $(0,1)$, the standard low-rank algorithm will output the factor $(0,1)$. Thus the optimal fair solution achieves value $\frac{1}{4}$ on the socially fair low-rank approximation objective while the standard low-rank approximation solution achieves value $\frac{1}{2}$, so that the ratio is a $50\%$ improvement.

