# OpenReview forum: "On Socially Fair Regression and Low-Rank Approximation"
_ICLR.cc/2024/Conference — Submitted to ICLR 2024_

### Official Review · Reviewer_5W2W · 2023-10-31

**Soundness:** 3 good
**Presentation:** 3 good
**Contribution:** 3 good
**Rating:** 6
**Confidence:** 3

**Summary:**

The paper studies the two problems of socially fair regression and socially fair low rank approximation (rank $k$). By "socially fair", it means the objective is to minimize the maximum value of loss over given 'protected' groups. The authors show simple polynomial time algorithms for fair regression however they show hardness results for even constant factor approximation to the fair LRA(Low rank approximation) problem. Under certain assumptions, the authors show a $2^{poly(k)}$ algorithm for the fair LRA problem. The authors use sketching techniques and several well-known results from randomized numerical linear algebra to achieve this. The authors support their theoretical claims with a set of proof -of -concept empirical results.

**Strengths:**

1) Regression and Low rank approximation are fundamental problems in Machine Learning and fairness is becoming a very essential aspect   of ML. As such the problems are important and will be of interest to the community.
2) The algorithms for fair regression are very simple and intuitive.
3) Though, according to me, the main and most technical contribution of the paper is theoretical in nature i.e., complexity analysis of the fair LRA problem, the authors have provided small set of experiments for the fair regression problem which allows a simple algorithm. The experiments complement well with the simple algorithms.
4) The paper, for the most part, is written clearly and appears sound (only had a high-level look at the proofs). Related work section is quite thorough.

**Weaknesses:**

1) The paper uses many existing results from randomized numerical linear algebra and other literature for the main technical results. Although I don't consider this by itself a weakness (in fact the ideas are combined nicely), it does have the unintended effect of making the paper less accessible to readers unfamiliar with the areas. It would be helpful if the writeup following theorems 1.5 and 1.6 is modified to make it more accessible.

**Questions:**

1. The question might sound a little basic. However, correct me if I am wrong, when you talk about the exponential time, you are only talking about time in terms of $k$ and not $n$ right? If yes please make it clearer, else it may create confusion regarding theorems 1.4 and 1.5.

2. What is the run time of the algorithm for bicriteria approximation in terms of $k$?
3. I believe here the groups are non-overlapping. Can you comment on the results were the groups overlapping?
4. What is the significance of the $\Delta$ in theorems 1.1. and 1.2. Is it guaranteed that the optimal $\mathbf{x}$ will also lie in the range given by $\Delta$?

My score is weak accept due to some of the confusion. I will be happy to raise the score if the above questions are answered satisfactoritly.

---

> ### Author Response · Authors · 2023-11-18
> **Response to Reviewer 5W2W**
>
> > The paper uses many existing results from randomized numerical linear algebra and other literature for the main technical results. Although I don't consider this by itself a weakness (in fact the ideas are combined nicely), it does have the unintended effect of making the paper less accessible to readers unfamiliar with the areas. It would be helpful if the writeup following theorems 1.5 and 1.6 is modified to make it more accessible.
>
> We apologize that the previous version may have been inaccessible to general audiences. We have added a new Appendix A.1 that describes high-level intuition for a number of common techniques, to provide additional guiding exposition intuition for the statements following Theorems 1.5 and 1.6, which we intend to incorporate into the main body in a full version of the paper.
>
> > The question might sound a little basic. However, correct me if I am wrong, when you talk about the exponential time, you are only talking about time in terms of $k$ and not $n$ right? If yes please make it clearer, else it may create confusion regarding theorems 1.4 and 1.5.
>
> Yes, we are referring to exponential time in $k$. We have updated the discussion surrounding Theorems 1.4 and 1.5 in the revised version to clarify this point.
>
> > What is the run time of the algorithm for bicriteria approximation in terms of $k$?
>
> The runtime for the bicriteria approximation algorithm is polynomial in $n$. Since $k\le n$, the runtime for the bicriteria approximation is also polynomial in $k$. In particular, there is a setting of the constants in the trade-offs so that the accuracy is maintained up to asymptotic factors and the runtime of the bicriteria algorithm is $O(nd^2+n^2d)$ just using naive matrix multiplication methods.
>
> > I believe here the groups are non-overlapping. Can you comment on the results were the groups overlapping?
>
> In general, socially fair algorithms do not seem well-defined when the groups are overlapping, i.e., if an individual can belong to multiple sub-populations. However, we agree with the reviewer that it could make sense for an individual to be able to contribute to multiple groups through some fractional weighting mechanism, which would be an interesting direction to explore.
>
> > What is the significance of the $\Delta$ in theorems 1.1. and 1.2. Is it guaranteed that the optimal $x$ will also lie in the range given by $\Delta$?
>
> The significance of $\Delta$ in Theorems 1.1 and 1.2 is the size of the space in each dimension, which is related to the size of the overall search space by our algorithm, and thus the overall runtime by our algorithm. We can choose a sufficiently large $\Delta$ to guarantee that the optimal $x$ will lie in the search space, given known bounds on the entries of the input matrices. We note that it is standard to assume bounds on the input matrices, since otherwise, the optimal solution can be arbitrarily large or small.

---

> > ### Comment · Reviewer_5W2W · 2023-11-21
> > **Thanks for the response**
> >
> > Thank you for your reply.
> >
> > I still feel there needs to be more clarity in terms of implications of theorems 1.3,1.4 and 1.5. At present they give a slightly confusing picture to me. Currently I will keep my score.

---

> > > ### Author Response · Authors · 2023-11-22
> > >
> > > Thanks for the input. We've added surrounding text and uploaded a new version that informally summarizes the relationship between Theorems 1.3, 1.4, and 1.5 as "Theorems 1.3 and 1.4 provide strong evidence for why Theorem 1.5 needs to use runtime $2^{\Omega(k)}$"

---

### Official Review · Reviewer_wDtZ · 2023-11-01

**Soundness:** 2 fair
**Presentation:** 1 poor
**Contribution:** 2 fair
**Rating:** 3
**Confidence:** 2

**Summary:**

This paper analyzed fair regression and fair low-rank approximation theoretically with an empirical evaluation of their proposed method of fair regression. The authors show by theorems that the low-rank approximation has a much larger computation complexity than the regression under fairness requirements. They propose two algorithms for approximation solutions for fair low-rank approximation.

**Strengths:**

The theoretical results for the fair low-rank approximation look novel and valuable to me.

The difference between the computation complexity of fair regression and other fair machine learning tasks (in this paper, the low-rank approximation) is interesting.

**Weaknesses:**

1. There is no real-world application or numerical evaluation of the proposed fair low-rank approximation method, which I think is more important than the evaluation of the fair regression method shown in Section 4.
2. There is no comparison between the proposed methods and other methods in the existing literature. For example, for fair regression, the authors may want to compare their convex solver method to Abernethy et al. (2022).
3. The contribution to socially fair regression seems limited since the method is simple and based on the observation in Abernethy et al. (2022).
4. This paper is not well-organized. Section 1.1 about the contributions is overwhelmingly long (about 2.5 pages), most of which is about the fair low-rank approximation which I think can be put in Section 3, especially since some parts in Section 1.1 and Section 3 are overlapped.

Some typos:
1. Page 4. 'Thus we consider the multi-response regression problem $\min_{B^{(i)}} \max_{i\in[\ell]}$ ...' I don't understand the $B^{(i)}$ since $i$ in used in the $\max$.
2. Theorem 1.5. $\delta$ is not defined.
3. Page 26. 'paragraphSynthetic dataset' looks strange to me.
4. In Algorithm 4 Line 2. It is unclear where S is generated from.

**Questions:**

1. For the approximate fair low-rank approximation, is it possible to have corresponding lower bounds for Theorems 1.5 and 1.6?
2. Can we replace the probability value 2/3 in Theorems 1.5 and 1.6 with some value $\xi$ which can be as close to 1 as possible or converge to 1 w.r.t. n? Otherwise, it seems like finding the approximate solution is still not guaranteed with high confidence.
3. The equation $\max \\|x\\|_\infty = (1\pm \epsilon)\\|x\\|_p$ in the beginning of Section 3.2 is surprising to me. Is there a typo?

---

> ### Author Response · Authors · 2023-11-18
> **Response to Reviewer wDtZ**
>
> > There is no real-world application or numerical evaluation of the proposed fair low-rank approximation method, which I think is more important than the evaluation of the fair regression method shown in Section 4.
>
> Our lower bounds show that under a common hardness conjecture, i.e., Exponential Time Hypothesis, exponential time is required for socially fair low-rank approximation algorithms. Thus, it does not seem feasible to implement proposed algorithms on large-scale real-world datasets. On the other hand, in the appendix, we provide experiments on synthetic datasets show
>
> > There is no comparison between the proposed methods and other methods in the existing literature. For example, for fair regression, the authors may want to compare their convex solver method to Abernethy et al. (2022). The contribution to socially fair regression seems limited since the method is simple and based on the observation in Abernethy et al. (2022).
>
> Yes, we remark that for socially fair regression, the main observation is that the problem is convex, similar to Abernethy et. al. (2022). Whereas Abernethy then uses a different convex solver method, we reference the interior point methods for a full end-to-end theoretical guarantee. On the other hand, for our experiments, we use the standard convex solver packages in Python. We believe empirical comparisons between the standard packages and the convex solvers of Abernethy et. al. (2022) are more a question about convex optimization benchmarks rather than regression benchmarks.
>
> In any case, we do not view the socially fair regression as our main result, but rather a simple warm-up that in conjunction our main socially fair low-rank approximation results, exhibits a somewhat surprising juxtaposition for the complexities of the two problems, given the abundance of shared techniques often used to approach these problems.
>
> > This paper is not well-organized. Section 1.1 about the contributions is overwhelmingly long (about 2.5 pages), most of which is about the fair low-rank approximation which I think can be put in Section 3
>
> We remark that the purpose of Section 1.1 is both to highlight our contributions but also to provide a technical overview/intuition for our results. We have renamed Section 1.1 to emphasize this intention.
>
> > some typos
>
> Thanks for pointing out this typos. We have addressed these points in the revised version.
>
> > For the approximate fair low-rank approximation, is it possible to have corresponding lower bounds for Theorems 1.5 and 1.6?
>
> Note that in some sense, Theorem 1.3 and Theorem 1.4 are lower bounds that correspond to Theorems 1.5 and 1.6. Theorem 1.3 notes that any one-sided constant-factor approximation in polynomial is unlikely under the assumption that P!=NP, which motivates the study of the bicriteria approximation in Theorem 1.6. Similarly, Theorem 1.4 shows that under the stronger Exponential Time Hypothesis, runtime that is exponential in $k$ is necessary, and Theorem 1.5 achieves runtime that has exponential dependnecies on $k$ (though admittedly not linear in the exponential). We agree that it would be an interesting question to resolve this remaining gap.
>
> > Can we replace the probability value 2/3 in Theorems 1.5 and 1.6 with some value $\xi$ which can be as close to 1 as possible or converge to 1 w.r.t. n? Otherwise, it seems like finding the approximate solution is still not guaranteed with high confidence.
>
> Yes, the success probability can be boosted to an arbitrarily high $1-\delta$ by running $O\left(\log\frac{1}{\delta}\right)$ independent instances of the algorithm and taking the best solution. In this case, the runtime will also increase by a factor of $O\left(\log\frac{1}{\delta}\right)$.
>
> > The equation $\max\|x\|_\infty=(1\pm\epsilon)\|x\|_p$ in the beginning of Section 3.2 is surprising to me. Is there a typo?
>
> The maximum is somewhat redundant, but the reason that the $L_\infty$ norm is close to the $L_p$ for large $p$ is that the $p$-th power of the large entries are significantly larger than the $p$-th power of the small entries, and thus $\|x\|_\infty=(1\pm\epsilon)\|x\|_p$ for $p=O\left(\log\frac{n}{\epsilon}\right)$, where $n$ is the dimension of $x$. See the proof of Lemma C.10 in the appendix for a formal proof.

---

> > ### Comment · Reviewer_wDtZ · 2023-11-19
> > **Thank you for your response**
> >
> > For Lemma C.10, I found it was necessary to make the statement clear: For the condition you give, $p=O(\frac{1}{\epsilon}\log \ell)$, it is equivalent to say that there exists $C$ such that $p\leq \frac{C}{\epsilon} \log\ell$ which means $\log\ell \geq \frac{p\epsilon}{C}$. We know that for fixed $C$, if $\epsilon$ is sufficiently large, $\frac{p\epsilon}{C} > p\log(1+\epsilon)$. Then, we have $\ell^{1/p} > (1+\epsilon)$ which contradicts your statement in Lemma C.10.
> >
> > Theorem 1.3 and 1.4 cannot be treated as a lower bound for Theorem 1.5 and 1.6 since they have different goals, (and the lower bound is larger than the upper bound.)
> >
> > I still think that it is hard to evaluate the contribution of this work based on its current results. Therefore, I will keep my rating score as it is now.

---

> ### Author Response · Authors · 2023-11-19
>
> Thanks for the feedback!
>
> Note that Section 3.2 assumes $\varepsilon\in(0,1)$, which crucially uses this fact in writing $\log(1+\varepsilon)=O(\varepsilon)$. We have clarified this in the statement and proof of Lemma C.10.
>
> We agree that Theorem 1.3 and Theorem 1.4 cannot be treated as pure lower bounds for Theorem 1.5 and Theorem 1.6 and we apologize for any such implications. However, we remark that Theorem 1.3 and Theorem 1.4 provides strong evidence that we cannot achieve even an approximation to socially fair low-rank approximation in runtime exponential in $k$, and thus we require additional relaxations, such as runtime exponential in $\text{poly}(k)$, i.e., Theorem 1.5, or a bicriteria approximation, i.e., Theorem 1.6.

---

### Official Review · Reviewer_2ktH · 2023-11-09

**Soundness:** 3 good
**Presentation:** 2 fair
**Contribution:** 2 fair
**Rating:** 6
**Confidence:** 2

**Summary:**

The paper solves socially fair regression and low-rank approximation problems. It shows that fair regression can be solved up to some accuracy in polynomial time. A constant factor approximation of a fair low-rank approximation problem requires longer time that exponentially depends on k.

**Strengths:**

The paper shows that an additive error approximation can be achieved in poly(n,d,eps^(-1)) for Lp regression where p in (1, infinity). It also gives an nnz(A) running time algorithm that ensures a relative error approximation of the problem.

It shows that getting a constant factor low-rank approximation for the socially fair low-rank problem is np-hard. Howeve, the running time reduces drastically when the approximation factors are functions of poly(l,k).

**Weaknesses:**

See questions.

**Questions:**

k-means clustering can also be viewed as constraint low-rank approximations. Can the socially fair low-rank approximation be used to solve the socially fair k-means clustering problem?

Notations are not well defined and difficult to follow, eg alpha is well defined in the context where ever it has been used.

---

> ### Author Response · Authors · 2023-11-18
> **Response to Reviewer 2ktH**
>
> > k-means clustering can also be viewed as constraint low-rank approximations. Can the socially fair low-rank approximation be used to solve the socially fair k-means clustering problem?
>
> Although the view of $k$-means clustering as a constrained low-rank approximation problem has been useful in results such as dimensionality reduction, it is not immediately clear to us how to use the socially fair low-rank approximation problem to solve the socially fair $k$-means clustering problem, in part due to the fact that the linear combination of the low-rank factors can be both fractional and negative. However, we do think that in general the relationships between $k$-means clusteirng and low-rank approximation are interesting to explore.
>
> > Notations are not well defined and difficult to follow, eg alpha is well defined in the context where ever it has been used.
>
> We first introduce and define $\alpha$ in the discussion after Theorem 1.5. We again introduce and define $\alpha$ is the discussion at the beginning of Section 3.1. Finally, we define $\alpha$ in Line 1 of Algorithm 4. Regardless, we have made a pass to clarify notations and variables, including repeating the definition of a variable if it has not been recently discussed in the paper. We hope this improves reader accessibility.

---

### Official Review · Reviewer_gYDC · 2023-11-10

**Soundness:** 2 fair
**Presentation:** 2 fair
**Contribution:** 2 fair
**Rating:** 3
**Confidence:** 4

**Summary:**

This paper studies two fair learning problems.

**Strengths:**

The paper has its own technical merits.

**Weaknesses:**

There are two major flaws:

First of all, the fair regression notion is very different from literature. The referee is afraid that this notion may not be meaningful at all. Since each group is more desired to match the prediction values rather than the losses. Please carefully go over the literature on fair regression.

Second, the low-rank approximation has not too much to do with fair regression. Piling two topics into one paper is making referee wondering that the contributions of this paper are insufficient on both ends.

With that, the referee recommends a rejection.

**Questions:**

There are two major flaws:

First of all, the fair regression notion is very different from literature. The referee is afraid that this notion may not be meaningful at all. Since each group is more desired to match the prediction values rather than the losses. Please carefully go over the literature on fair regression.

Second, the low-rank approximation has not too much to do with fair regression. Piling two topics into one paper is making referee wondering that the contributions of this paper are insufficient on both ends.

With that, the referee recommends a rejection.

---

> ### Author Response · Authors · 2023-11-18
> **Response to Reviewer gYDC**
>
> > First of all, the fair regression notion is very different from literature. The referee is afraid that this notion may not be meaningful at all. Since each group is more desired to match the prediction values rather than the losses. Please carefully go over the literature on fair regression.
>
> We respectfully disagree with the notion that the fair regression notion is very different from literature. Although there are many possible definitions for fairness, the notion of social fairness (also called min-max fairness) is well-studied. For example, in Section 1.2 on related works, we refer to various literature that study social fairness on principal component analysis/low-rank approximation [STM+18,TSS+19], regression [AAM+22], and clustering [GSV21,ABV21,MV21,CMV22].
>
> [STM+18] Samira Samadi, Uthaipon Tao Tantipongpipat, Jamie Morgenstern, Mohit Singh, Santosh S. Vempala: The Price of Fair PCA: One Extra dimension. NeurIPS 2018: 10999-11010
>
> [TSS+19] Uthaipon Tantipongpipat, Samira Samadi, Mohit Singh, Jamie Morgenstern, Santosh S. Vempala: Multi-Criteria Dimensionality Reduction with Applications to Fairness. NeurIPS 2019: 15135-15145
>
> [GSV21] Mehrdad Ghadiri, Samira Samadi, Santosh S. Vempala: Socially Fair k-Means Clustering. FAccT 2021: 438-448
>
> [ABV21] Mohsen Abbasi, Aditya Bhaskara, Suresh Venkatasubramanian: Fair Clustering via Equitable Group Representations. FAccT 2021: 504-514
>
> [MV21] Yury Makarychev, Ali Vakilian: Approximation Algorithms for Socially Fair Clustering. COLT 2021: 3246-3264
>
> [CMV22] Eden Chlamtác, Yury Makarychev, Ali Vakilian: Approximating Fair Clustering with Cascaded Norm Objectives. SODA 2022: 2664-2683
>
> [AAM+22] Jacob D. Abernethy, Pranjal Awasthi, Matthäus Kleindessner, Jamie Morgenstern, Chris Russell, Jie Zhang:
> Active Sampling for Min-Max Fairness. ICML 2022: 53-65
>
> > Second, the low-rank approximation has not too much to do with fair regression. Piling two topics into one paper is making referee wondering that the contributions of this paper are insufficient on both ends.
>
> Regression and low-rank approximation (including column subset selection) are often studied together in approximation algorithms, because there is often a set of common techniques that can be applied to both problems. For example, [Woo14] showed that the popular "sketch-and-solve" approach can be applied to both problems by generating a random Gaussian matrix of appropriate dimension to apply dimensionality reduction and then solving the problem in the reduced space. Similarly, it is known that the related leverage score sampling and ridge leverage score sampling [CMM17] techniques can be applied to regression and low-rank approximation respectively. Moreover, there exist online versions of these techniques for the streaming model [BDM+20,CMP20], further relating the complexity of these two problems.
>
> Thus as highlighted in Section 1.1, the main contribution of this paper is an arguably surprising message that for the problem of social fairness, linear regression and low-rank approximation exhibit drastically different behaviors. We show that this is because socially fair regression can be cast as a convex optimization problem while socially fair low-rank approximation has intrinsic combinatorial properties that allow it to compute the minimal distance between a set of $n$ points and a $k$-dimensional subspace, which is known to be a "hard" problem.
>
> [Woo14] David P. Woodruff: Sketching as a Tool for Numerical Linear Algebra. Found. Trends Theor. Comput. Sci. 10(1-2): 1-157 (2014)
>
> [CMM17] Michael B. Cohen, Cameron Musco, Christopher Musco: Input Sparsity Time Low-rank Approximation via Ridge Leverage Score Sampling. SODA 2017: 1758-1777
>
> [CMP20] Michael B. Cohen, Cameron Musco, Jakub Pachocki: Online Row Sampling. Theory Comput. 16: 1-25 (2020)
>
> [BDM+20] Vladimir Braverman, Petros Drineas, Cameron Musco, Christopher Musco, Jalaj Upadhyay, David P. Woodruff, Samson Zhou: Near Optimal Linear Algebra in the Online and Sliding Window Models. FOCS 2020: 517-528

---

### Author Response · Authors · 2023-11-18
**Thanks to all Reviewers**

We thank the reviewers for their thoughtful comments and valuable feedback. We appreciate the positive remarks, such as
* The paper has its own technical merits. (Reviewer gYDC)
* The theoretical results for the fair low-rank approximation look novel and valuable to me. (Reviewer wDtZ)
* The difference between the computation complexity of fair regression and other fair machine learning tasks (in this paper, the low-rank approximation) is interesting. (Reviewer wDtZ)
* Regression and low rank approximation are fundamental problems in machine learning and fairness is becoming a very essential aspect of ML. As such the problems are important and will be of interest to the community. (Reviewer 5W2W)
* The algorithms for fair regression are very simple and intuitive. (Reviewer 5W2W)
* Though, according to me, the main and most technical contribution of the paper is theoretical in nature i.e., complexity analysis of the fair LRA problem, the authors have provided small set of experiments for the fair regression problem which allows a simple algorithm. (Reviewer 5W2W)
* The experiments complement well with the simple algorithms. (Reviewer 5W2W)
* The paper, for the most part, is written clearly and appears sound. (Reviewer 5W2W)
* Related work section is quite thorough. (Reviewer 5W2W)

We have incorporated reviewer feedback into an updated version of the document to further improve overall presentation -- **we have uploaded this new version of the paper** and we have **highlighted our changes in blue**. In particular, we have added significant discussion in Appendix A.1 summarizing a number of existing unifying techniques that can be used for both linear regression and low-rank approximation, emphasizing the surprising message of our paper that socially fair linear regression and socially fair low-rank approximation exhibit two vastly different complexities.

More generally, we provide our responses to the initial comments of each reviewer below. We hope our answers resolve all initial questions and concerns raised by the reviewers and we will be most happy to answer any remaining questions during the discussion phase!

---

> ### Author Response · Authors · 2023-11-21
>
> As a follow-up, we have further clarified the relationships between Theorems 1.3, 1.4, 1.5, and 1.6 in the revised uploaded version. Specifically, we note:
> > Together, Theorem 1.3 and Theorem 1.4 show that under standard complexity assumptions, we cannot achieve constant-factor approximation to fair low-rank approximation using time polynomial in $n$ and exponential in $k$. We thus consider additional relaxations, such as bicriteria approximation (Theorem 1.6) or $2^{\text{poly}(k)}$ runtime (Theorem 1.5).
>
> We emphasize that Theorem 1.3 and Theorem 1.4 are not meant to be matching lower bounds with Theorem 1.5 and Theorem 1.6, but rather strong evidence as to why additional relaxations, such as those in Theorem 1.5 and Theorem 1.6, are necessary.

---

### Meta-Review · Area_Chair_UKZK · 2023-12-12

**Metareview:**

Despite author responses, there remain concerns on the exposition of the contributions of the work.

**Justification For Why Not Higher Score:**

Most reviewers were negative and thought the paper was organized poorly to convey the main contributions.

**Justification For Why Not Lower Score:**

N/A

---

### Decision · Program_Chairs · 2024-01-16

Reject